# Ice nucleation by smectites: The role of the edges

Anand Kumar[1,2], Kristian Klumpp[2], Chen Barak[3], Giora Rytwo[3,4], Michael Plötze[5], Thomas Peter[2], Claudia Marcolli[2]

[1]Department of Mechanical Engineering, The University of British Columbia, Vancouver V6T1Z1, Canada

[2]Institute for Atmospheric and Climate Sciences, ETH Zurich, Zurich, 8092, Switzerland

[3]Environmental Physical Chemistry Laboratory, MIGAL Galilee Research Center, Kiryat Shmona 1101600, Israel

[4]Environmental & Water Sciences Departments, Tel Hai College, Upper Galilee 1220800, Israel

[5]Institute for Geotechnical Engineering, ETH Zurich, Zurich 8093, Switzerland

*Correspondence to*: claudia.marcolli@env.ethz.ch; anand.kumar@ubc.ca

## Abstract

Smectites, like other clay minerals, have been shown to promote ice nucleation in the immersion freezing mode and likely contribute to the population of ice nucleating particles (INPs) in the atmosphere. Smectites are layered aluminosilicates, which form platelets that depending on composition might swell or even delaminate in water by intercalation of water molecules between their layers. They comprise among others montmorillonites, hectorites, beidellites, and nontronites. In this study, we investigate the ice nucleation (IN) activity of a variety of natural and synthetic smectite samples with different exchangeable cations. The montmorillonites STx-1b and SAz-1, the nontronite SWa-1, and the hectorite SHCa-1 are all rich in $Ca^{2+}$ as the exchangeable cation, the bentonite MX-80 is rich in $Na^+$ with a minor contribution of $Ca^{2+}$, the synthetic Laponite is a pure $Na^+$ smectite. The bentonite SAu-1 is rich in $Mg^{2+}$ with a minor contribution of $Na^+$, and the synthetic interstratified mica-montmorillonite Barasym carries $NH_4^+$ as the exchangeable cation. In emulsion freezing experiments, all samples except Laponite exhibited one or two heterogeneous freezing peaks with onsets between 239 K and 248 K and a quite large variation in IN activity, yet without clear correlation with the exchangeable cation, with the type of smectite, or with mineralogical impurities in the samples. To further investigate the role of the exchangeable cation, we performed ion exchange experiments. Replacing $NH_4^+$ with $Ca^{2+}$ in Barasym reduced its IN activity to that of other Ca-rich montmorillonites. In contrast, stepwise exchange of the native cations in STx-1b once with $Y^{3+}$ and once with $Cu^{2+}$ showed no influence on IN activity. However, aging of smectite suspensions in pure water up to several months revealed a decrease in IN activity with time, which we attribute to the delamination of smectites in aqueous suspensions, which may proceed over long timescales. The dependence of IN activity on platelet stacking and thickness can be explained if the hydroxylated chains forming at the edges are the location of ice nucleation in smectites, since the edges need to be thick enough to host a critical ice embryo. We hypothesize that at least three smectite layers need to be stacked together to host a critical ice embryo on clay mineral edges and that the larger the surface edge area is the higher the freezing temperature. Comparison with reported platelet thicknesses of the investigated smectite samples suggests that the observed freezing temperatures are indeed limited by the surface area provided

by the mostly very thin platelets. Specifically, Laponite, which did not show any IN activity, is known to delaminate into single layers of about 1 nm thickness, which would be too thin to host a critical ice embryo.

## 1 Introduction

A major unknown in Earth's climate change projections is the effect of changing cloud cover (Nazarenko et al., 2017), and how the presence of ice in clouds influences the radiative properties and lifetime of clouds, and precipitation (Shupe et al., 2008; Mülmenstädt et al., 2015; Heymsfield et al., 2020). In the absence of ice-nucleating particles (INPs) clouds may supercool to temperatures around 236–238 K, where homogeneous ice nucleation sets in and leads to cloud glaciation (Ickes et al., 2015; Lohmann et al., 2016), while at temperatures between the homogeneous freezing threshold and the ice melting temperature, freezing is initiated by INPs, which contain surfaces that are able to promote ice nucleation (Pruppacher and Klett, 1994; Vali et al., 2015). There is increasing evidence, e.g. from refreeze experiments, that the ability to nucleate ice is restricted to few nucleation sites on the surface of an INP, while the largest part of it remains ice nucleation (IN) inactive (Wright and Petters, 2013; Kaufmann et al., 2017; Holden et al., 2021). Yet, not all IN sites have the same activity. Based on classical nucleation theory (CNT), each nucleation site has a temperature range of a few Kelvin, in which its ice nucleation rate increases from hardly relevant to highly probable (Vali, 2014; Kaufmann et al., 2017). Within the framework of CNT, the IN activity of dust samples have successfully been described by a lognormal distribution of contact angles (Marcolli et al., 2007; Lüönd et al., 2010; Niedermeier et al., 2011; 2014), such that average IN sites are frequent and best sites are rare (Pinti et al., 2012). Due to this strong temperature dependence, the singular hypothesis seems justifiable, as it assumes a characteristic temperature (Vali, 2008), at which INPs induce freezing deterministically. While this assumption is simplistic, it represents the freezing behavior of an ensemble of INPs better than the stochastic hypothesis, which conjectures identical INPs that induce freezing with a temperature-dependent nucleation rate (Vali et al., 2014; Knopf et al., 2020). To improve our understanding of heterogeneous ice nucleation, it is indispensable to learn more about the special features that discriminate nucleation sites from the rest of the INP surfaces. While there is increasing evidence that pores are the key feature required for ice nucleation occurring below water saturation (Marcolli, 2014; 2020; David et al., 2019; Campbell et al., 2017; 2018), much less is known about ice nucleation occurring via immersion freezing, i.e. when the INP is immersed in aqueous droplets (Vali et al., 2015). One way to improve our understanding is to analyze and compare the differences and similarities between the IN activities of structurally similar INPs.

Mineral dust has been found to be the prevailing INP type at temperatures below 258 K (Kanji et al., 2017; O'Sullivan et al., 2018; Brunner et al., 2021), while biological particles such as bacteria, pollen, fungi, and proteins are considered to dominate ice nucleation at higher temperatures (DeMott and Prenni, 2010; Després et al., 2012; Kanji et al., 2017). Feldspar, quartz, and clay minerals have proven to be relevant mineral INPs in the immersion freezing mode (Atkinson et al., 2013; Pinti et al., 2012; Kanji et al., 2017; Kaufmann et al., 2016; Boose et al., 2016; Kumar et al., 2018; 2019a; 2019b). K-feldspars such as microcline attracted special attention as they induce freezing at the highest temperatures among mineral dust INPs (Aktinson

et al., 2013; Kiselev et al., 2016) with freezing onsets around 251–252 K in emulsion freezing experiments (Kumar et al., 2018; Klumpp et al., 2022). Yet, they also proved to be highly sensitive to the presence of solutes, as the IN activity of feldspar suspensions in aqueous salt solutions was found to decline for solute concentrations >0.02 M (Whale et al., 2018; Kumar et al., 2018; Kumar et al., 2019b; Yun et al., 2020; 2021). Moreover, they irreversibly lose IN activity in the presence of acidic solutes (Kumar et al., 2018; Klumpp et al., 2022). Conversely, the presence of ammonia or ammonium containing salts boosts their IN activity above the pure water case (Kumar et al., 2018; 2019b; Whale et al., 2018; Worthy et al., 2021).

Clay minerals are also IN active, albeit at lower temperatures than K-feldspars (Pinti et al., 2012; Kanji et al., 2017). As they belong to the fine particle fraction, they become enriched in aerosols during long-range transport (Murray et al., 2012), which enhances their atmospheric relevance compared with quartz and feldspars, which belong to the coarse particle fraction. Feldspars and clay minerals are both aluminosilicates, although with fundamentally different structures. Feldspars consist of a three-dimensional aluminosilicate tetrahedral network. On the other hand, clay minerals constitute the phyllosilicate or sheet silicate family of minerals, which are characterized by their layered structures composed of sheets of $SiO_4$ tetrahedra (T) attached to octahedra (O) sheets (containing aluminum, magnesium, iron). Clay minerals are formed as a result of chemical weathering of other silicate minerals such as feldspars, volcanic glass, micas and ferromagnesian phases (Drever, 1997; Langmuir, 1997; Christidis and Huff, 2009). Major subgroups of clay minerals include kandite (T—O layers), illite (T—O—T layers), smectite (T—O—T layers) and vermiculite (T—O—T layers). Because of this layered structure, clay mineral particles are anisotropic and exhibit different types of surfaces, namely basal siloxane and alumina surfaces and edges. In several clay minerals, due to isomorphic substitution of Si(IV) by Al(III) in the T-layer and Al(III) by Mg(II) in the O-layer, the basal surfaces may carry permanent negative charge that is balanced by exchangeable cations such as $Na^+$, $Ca^{2+}$, $Mg^{2+}$ or $K^+$ in the interlayer space and on the surface (Bleam, 2017). Clay mineral edges consist of hydroxylated Al–O–Si–OH structures, with pH dependent charge sites (Macht et al., 2011).

Among clay minerals, illites, and specifically illite NX from the Clay Minerals Society (CMS) received attention as a proxy for atmospheric mineral dust (Broadley et al., 2012; Hiranuma et al., 2015; Garimella et al., 2016). Illite layers consist of an octahedral sheet sandwiched by two tetrahedral sheets with a high degree of isomorphic substitution. They possess two basal siloxane surfaces with permanent negative charge, which is predominantly balanced by $K^+$ (Nieto et al., 2010; Shao et al., 2019). Illite has been referred to as K-deficient mica (Gualtieri and Ferrari, 2006) with small crystal size, poor crystallinity, non-swelling and low mineralogical purity (Rieder et al., 1998; Nieto et al., 2010). In emulsion freezing experiments, illite NX exhibited onset freezing temperatures around 245 K (Pinti et al., 2012).

Kaolinite is a 1:1 layered aluminosilicate with each layer composed of an O-sheet connected through bridging oxygens to a T-sheet. Thus, kaolinite particles consist of platelets with two different basal surfaces i.e. a siloxane and an aluminum hydroxide basal surface, in addition to the hydroxylated edges. As there is little isomorphic substitution, there are only few exchangeable cations required to balance the charge (Bickmore et al., 2002). In contrast to illite, kaolinite can be found mineralogically almost pure. CMS provides two kaolinite samples, KGa-1b and KGa-2, which both exhibit a high mineralogical purity of 96 %, yet, differ in crystallinity (Chipera and Bish, 2001). In emulsion freezing experiments, KGa-1b and KGa-2 both exhibit

onset freezing temperatures around 240–242 K (Pinti et al., 2012). On the other hand, freezing onset temperatures up to 245
K are reached by halloysites, which are chemically almost identical to kaolinites but typically occur in the form of cylindrical
tubes (Churchman et al., 1995; Joussein et al., 2005; Pasbakhsh et al., 2013; Klumpp et al., 2023). Immersion freezing
experiments with single particles revealed that not all kaolinite particles are IN active and that the fraction of IN inactive
particles increases with decreasing particle size, in accordance with the assumption that ice nucleation is restricted to rare
nucleation sites (Lüönd et al., 2010; Welti et al., 2012; Wex et. 2014; Nagare et al., 2016). Interestingly, kaolinite and halloysite
showed an increased IN activity in the presence of ammonia or ammonium solutes (Kumar et al., 2019b, Klumpp et al., 2023),
pointing to a similar chemical makeup of nucleation sites as feldspars.

Smectites stand out from the other members of the clay family due to their unique property to accommodate high degree of
interlayer molecules, including water and organics. This is termed as swelling property. In smectites each layer consists of one
O-sheet sandwiched by two T-sheets. Through isomorphic substitution, negative charge is introduced in the layers that is
balanced by exchangeable cations, most frequently $Na^+$ and/or $Ca^{2+}$. In montmorillonites, isomorphic substitution is
predominantly or even exclusively present in the O-layer, where Al(III) is typically replaced by Mg(II). Beidellites, on the
other hand, are characterized by isomorphic substitution in the tetrahedral sheet, where Si(IV) is typically replaced by Al(III).
When in addition to the substitution of Si(IV) by Al(III) in the T-sites, Al(III) is substituted by Fe(III) in the O-sites, the
smectite is referred to as nontronite. Smectites can be further subdivided in dioctahedral species such as montmorillonites or
beidellites with $[Al_2Si_4O_{10}(OH)_2]$ sandwiched between the tetrahedral sheets and in trioctahedral species, e.g. hectorites with a
$[Mg_3Si_4O_{10}(OH)_2]$ (talc) sheet instead. Smectite particles consist of platelets with two basal siloxane surfaces and the
hydroxylated edges with pH dependent OH groups. Smectite layers are randomly rotated with respect to each other (i.e. they
are turbostratic) and, due to the charge balancing ions that form outer sphere complexes to the siloxane surfaces, they are
swelling (Mystkowski et al., 2000). When $Ca^{2+}$ is the prevailing charge-balancing cation, the swelling is limited to three to
four water layers intercalated between the smectite sheets. If $Na^+$ is intercalated between the smectite layers, swelling can lead
to complete delamination of the smectite sheets (Cases et al., 1992; Metz et al., 2005; Segad et al., 2012).

Pinti et al. (2012) measured the IN activity of four different montmorillonites, K-10 and KSF from Sigma Aldrich, and STx-
1b and SWy-2 from CMS in emulsion freezing experiments. The montmorillonites from Sigma Aldrich were both acid treated
leading to partial disruption (K-10) and even complete delamination of the sheets. While the natural montmorillonites from
CMS exhibited a heterogeneous freezing peak already at 0.1 wt % suspension concentration, the two montmorillonites from
Sigma Aldrich required suspension concentrations of 2 wt % (K-10) or even 5 wt % (KSF) to show a heterogeneous freezing
signal. The montmorillonites STx-1b and the ones from Sigma Aldrich had freezing onsets at around 240 K in emulsion
freezing experiments, while SWy-2 showed a second freezing peak with onset around 247 K.

In this study, we have extended the emulsion freezing experiments performed by Pinti et al. (2012) to other smectite samples,
including natural and synthesized hectorites and beidellites. To identify the role of the exchangeable cations in smectites, we
performed ion exchange experiments and compared the IN activity before and after ion exchange. Finally, we use the gained
information to identify the most likely location of ice nucleation in smectites and clay minerals in general.

## 2 Methodology

### 2.1 Sample sources and mineralogy

We used six different naturally occurring smectite samples, most of them procured from Clay Minerals Society (CMS). All samples were obtained in form of fine powders. Using X-ray diffraction (XRD), detailed mineralogical compositions of the samples were obtained. Table 1 shows the mineralogical composition of all samples. XRD results showed that out of the six samples, three samples were Na/Ca-rich montmorillonites. A bentonite (Na-rich), a hectorite and a Fe-containing smectite sample were also included. In addition, we investigated two synthetic clay minerals, namely, smectite Laponite and

mica/smectite Barasym. Similar to Laponite, Barasym is a synthetic mica/smectite albeit with only $NH_4^+$ ions acting as the charge balancing cations (Moll, 2001). The chemical composition is given in Table 2 and a physicochemical characterization in Table 3. Below we summarize the characteristics of the investigated smectites:

**STx-1b** is a Ca-rich montmorillonite from Texas available through CMS. Isomorphic substitution of Al(III) by Mg(II) in the O-layer introduces a negative charge of -0.68 per unit cell (Mermut and Lagaly, 2001). There is no isomorphic substitution in

the T-layer, thus, the interlayer charge equals -0.68, which is only partly balanced by interlayer cations resulting in an unbalanced charge of -0.08 per unit cell (Mermut and Lagaly, 2001). The sample consists of 70.2 wt % smectite, which is intergrown with Opal-CT as the major impurity (Sanders et al. 2010).

**SAz-1** is another international standard sample from CMS sourced from a deposit in Arizona, United States of America (USA). Similar to STx-1b, it is a Ca-rich montmorillonite that only bears octahedral charge (-1.08) and has an unbalanced charge of

0.08 per unit cell. It has a high mineralogical purity (97.7 wt %) with only very minor shares of quartz and sanidine.

**SWa-1** is an iron-rich dioctahedral smectite (nontronite) from Grant County (Washington, USA). It bears an interlayer charge of -1.09 per unit cell that originates predominantly from isomorphous substitution in the T-layer (-0.91 per unit cell) (Mermut and Lagaly, 2001). It has a high mineralogical purity (92.1 wt %) with minor shares of quartz and calcite.

**SAu-1** is a bentonite of high purity with 10–20 wt % interstratified illite (Churchman et al., 2002; Gates, 2004) from the

Arumpo deposit (Australia) that has been recently added to the CMS Source Clay Repository. It is rich in $Mg^{2+}$ and $Na^+$ as the exchangeable cations (Churchman et al., 2002; Gates, 2004). The layer charge is distributed between T- (45 %) and O-sheets, and, most likely due to the interstratification with illite, it has a low swelling capability.

**SHCa-1** is a trioctahedral smectite (hectorite) with $Li^+$ substituting $Mg^{2+}$ in the O-sheet. It is sourced from a deposit in San Bernardino County (California, USA), and also available through CMS. It carries most of the negative charge in the O-sheet

(-1.35) and minor charge in the T-sheet (-0.22), resulting in an interlayer charge of -1.57 and 0.02 unbalanced charge per unit cell (Mermut and Lagaly, 2001). It contains 47.8 wt % hectorite together with a major calcite impurity (41.8 wt % in our sample), which may have a cementing effect on the hectorite (Stepkowska et al., 2004).

**Barasym** (SSM-100) is a synthetic mica-montmorillonite (NL Industries) with $NH_4^+$ as the exchangeable cation. It is available through CMS. Barasym carries most charge in the T-sheets (beidellite) and consists of irregularly mixed mica/smectite layers

at a ratio 2:1 (Stepkowska et al., 2004; Moll et al., 2001). As a synthetic clay, it is mineralogically almost pure with only a minor impurity of boehmite ($2.7 \pm 0.6$ wt %).

**Laponite RD** (Laponite® RD) is a synthetic trioctahedral smectite with $Li^+$ substituted for $Mg^{2+}$ in the O-sheet. It consists of a random mixture of hectorite, stevensite and kerolite layers with 40–50 % hectorite (Christidis et al., 2018). As it is a synthetic product, it is a mineralogically pure smectite. In suspension, it consists of monodispersed platelets with lateral layer dimensions

of about 30 nm, which are ~0.96 nm thick (i.e. one unit layer) (Delavernhe et al. 2018). It was provided by BYK-Additives Ltd, Widnes, Chesite, UK. An idealized structure of Laponite would have a neutral charge with six $Mg^{2+}$ in the O-layer, giving a positive charge of twelve. However, few $Mg^{2+}$ ions are substituted by $Li^+$ and some positions are empty to give a composition with typical empirical formula: $Na_{0.7}[(Si_8Mg_{5.5}Li_{0.3})O_{20}(OH)_4]$. There is a negative charge of 0.7 per unit cell, which is neutralized by $Na^+$ (Ghadiri et al., 2013; Zulian et al., 2014).

**MX-80** is a Wyoming montmorillonite that contains about 86 wt % montmorillonite together with minor shares of quartz, feldspar, cristobalite and pyrite. It was kindly supplied by Dr. Stefan Dultz (University of Hannover, Germany) (Dultz et al., 2005). The montmorillonite is rich in $Na^+$ with a minor share of $Ca^{2+}$. MX-80 is the commercial trade name for CMS source clays SWy-1/SWy-2/SWy-3, albeit with slight differences in composition from batch-to-batch.

### 2.2 X-ray diffraction

XRD measurements were made on randomly oriented and on textured powder specimens using a Bragg-Brentano diffractometer (Bruker AXS D8) using Co Kα radiation. The instrument was equipped with an automatic theta compensating divergence and antiscattering slit, primary and secondary soller slits and a Sol-X solid state detector. The qualitative phase composition was determined with the software DIFFRACplus (BRUKER AXS). On the basis of the peak position and their relative intensity the mineral phases were identified in comparison to the PDF-2 data base (International Centre for Diffraction

Data). For oriented specimens the basal reflexes of layer silicates are enhanced thereby facilitating their identification. A (001) peak of the smectite at ~1.5 nm indicates the presence of divalent cations in the interlayer while a peak at 1.2 nm is typical for monovalent cations. The changes in the reflex positions in the XRD pattern by intercalation of different organic compounds (e.g. ethylene glycol) and after heating were used for identification of smectite. Rietveld refinement using Profex software was performed on XRD patterns of randomly oriented specimens for a quantitative analysis (Doebelin and Kleeberg, 2015).

### 2.3 Sample treatment


STx-1b and Barasym samples were modified by performing a cation exchange procedure as described here. For convenience, the unaltered and the ion exchanged samples are referred to as "*original*" and "*modified*", respectively.


**Table 1.** Mineralogical composition of the investigated smectites in weight % ± 3 standard deviations (SD)

| Minerals | STx-1b | SAz-1 | SWa-1 | SAu-1 | SHCa-1 | Barasym | Laponite RD | MX-80 |
|---|---|---|---|---|---|---|---|---|
| Smectite | 70.2 ± 0.8 | 97.9 ± 0.3 | 92.1 ± 0.6 | 85.6 ± 0.7 | 47.8 ± 0.7 | 97.3 ± 4.5 | 100.0 | 85.8 ± 0.9 |
| Calcite | – | – | 1.5 ± 0.2 | 0.7 ± 0.1 | 41.8 ± 0.6 | – | – | – |
| Quartz | 0.9 ± 0.1 | 0.9 ± 0.1 | 6.4 ± 0.2 | 6.7 ± 0.2 | 2.6 ± 0.2 | – | – | 4.8 ± 0.2 |
| Dolomite | – | – | – | – | 2.0 ± 0.3 | – | – | – |
| Gypsum | – | – | – | 0.1 ± 0.1 | – | – | – | – |
| Analcime | – | – | – | – | 0.6 ± 0.1 | – | – | – |
| Na-Plagioclase | – | – | – | – | 5.3 ± 0.7 | – | – | – |
| Pyrite | – | – | – | 0.4 ± 0.1 | – | – | – | – |
| Opal CT | 21.4 ± 0.5 | – | – | – | – | – | – | – |
| K-Feldspar | 1.9 ± 0.3 | – | – | 0.7 ± 0.5 | – | – | – | 3.1 ± 0.4 |
| Sanidine | | – | – | 2.8 ± 0.5 | – | – | – | – |
| Kaolinite | 1.2 ± 0.4 | – | – | | – | – | – | – |
| Chlorite | | – | – | 1.6 ± 0.4 | – | – | – | – |
| Cristobalite | 2.3 ± 0.4 | – | – | 1.4 ± 0.2 | – | – | – | 6.0 ± 0.4 |
| Boehmite | – | – | – | – | – | 2.7 ± 0.6 | – | – |
| Sanidine | – | 1.2 ± 0.3 | – | – | – | – | – | – |
| Pyrite | – | – | – | – | – | – | – | 0.3 ± 0.1 |

### 2.3.1 STx-1b: Ion exchange and adsorption

The aim behind conducting the surface modification of STx-1b was to assess the influence of a bivalent ($Cu^{2+}$) and a trivalent ($Y^{3+}$) cation on the sample's IN ability. $Cu^{2+}$ adsorbs preferentially at the edge sites, and its binding affinity is larger than other divalent cations of similar ionic size, while trivalent cations in general are considered to bind to clays stronger than mono- or divalent cations (Undabeytia et al., 2005; Buzetzky et al., 2017)). We devised a step-wise cation exchange procedure to reach the cation exchange capacity (CEC) of 0.84 meq $g^{-1}$ of the sample (as reported by CMS). A stock suspension of 1 wt % STx-1b and another stock solution of 4 mmolal $YCl_3$ (Sigma Aldrich, 99 % purity) solution was prepared in Milli-Q water (resistivity ≥ 18.2 MΩcm). Twelve aliquots of 5 ml each were taken from the STx-1b suspension and placed in polypropylene tubes. Then appropriate aliquots of $YCl_3$ solution were added to each and diluted with Milli-Q water to bring the final volume of each sample to 10 ml. This led to a step-wise increasing concentration of $Y^{3+}$ (ranging from 0.11–2.67 mmolal) exposed to the same mass of STx-1b present in each tube. Each concentration step was prepared in triplicates. All samples were shaken on a laboratory-grade agitation table for 72 h to allow the mineral to equilibrate with the solution. Similarly, $Cu^{2+}$-exchanged STx-1b samples were prepared using 4 mmolal $CuCl_2$ (Sigma Aldrich, 99.9 % purity) solution. The $Cu^{2+}$ concentration ranged

from 0.16–3.33 mmolal. In addition, we prepared a reference blank sample (in triplicate) where STx-1b was only exposed to Milli-Q water.

After equilibration, aliquots from each set of triplicates were taken for particle charge measurement (see Sect. 2.3.2). The remaining samples were centrifuged at 3000 rpm for 1 h. The supernatant solution was collected for analysing the elemental composition (see Sect. 2.3.2). The settled particles (in form of slurry/paste) were flash frozen at -80 °C then kept in a lyophilizer freeze dryer (Christ Alpha 1-2 LD Plus, Germany) at -50 °C overnight. The freeze-drying process involves lowering the pressure and removing the ice from the sample by sublimation. Freeze drying was preferred over heat drying, as the sample is frozen in the former case which halts any further interaction between the particles and solution. The dried powder samples were homogenized using a mortar and pestle before conducting emulsion freezing experiments (Sect. 2.4).

**Table 2.** Chemical composition of the investigated smectites

| Smectite | Structure |
| --- | --- |
| STx-1b | $(Ca_{0.27}Na_{.0.04}K_{0.01})[Al_{2.41}Fe(III)_{0.09}Mn_{tr}Mg_{0.71}Ti_{0.03}][Si_{8.00}]O_{20}(OH)_4$ [a] |
| SAz-1 | $(Ca_{0.39} Na_{0.36}K_{0.02})[Al_{2.71}Mg_{1.11}Fe(III)_{0.12} Mn_{0.01}Ti_{0.03}][Si_{8.00}]O_{20}(OH)_4$ [a] |
| SWa-1 | $(Mg_{0.18}Ca_{0.36}K_{0.01})[Al_{0.61}Fe(III)_{3.08}Mn_{tr}Mg_{0.24}Ti_{0.07}][Si_{7.09}Al_{0.91}]O_{20}(OH)_4$ [a] |
| SAu-1 | $(Mg_{0.45}Na_{0.16}K_{0.035})[ Al_{2.84}Fe(III)_{0.45}Mg_{0.79}][Si_{7.62}Al_{0.38}]O_{20}(OH)_4$ |
| SHCa-1 | $(Mg_{0.56}Na_{0.42}K_{0.05})[Mg_{4.60}Li_{1.39}Mn_{tr}Ti_{0.01}][Si_{7.75}Al_{0.17}Fe(III)_{0.05}]O_{20}(OH)_4$ [a] |
| Barasym | $(Mg_{0.06} Na_{0.12}K_{tr}) [Al_{3.99}Fe(III)_{tr}Mn_{tr}Ti_{tr}][Si_{6.50}Al_{1.50}]O_{20}(OH)_4$ [a] |
| Laponite RD | $Na_{0.7}[Mg_{5.5}Li_{0..3}]Si_8O_{20}(OH)_4$ [b] |
| MX-80 | $M^+_{0.65}[Al_{3.11}Fe(III)_{0.36}Mg_{0.47}Ti_{0.01}][Si_{7.93}Al_{0.07}]O_{20}(OH)_4$ [c] |

[a] Clay Minerals Society
[b] Christidis et al. (2018)
[c] Segad et al. (2010)

### 2.3.2 STx1b: Particle charge and supernatant solution composition

*Particle charge.* As mentioned earlier, after reaching equilibrium, aliquots of suspensions were taken for charge measurements using a Particle Charge Detector (BTG Mütek GmbH PCD 03). The montmorillonite particles in water/solution carry negative charges. This leads to a concentration of oppositely charged ions (or counterions) on the particle surfaces. PCD is a polyelectrolyte titration method which uses a charged polyelectrolyte (of opposite charge compared to the sample) of known charge density (Rytwo et al., 2014; 2016). PCD allows shearing off of the counterions from the particle surfaces, leading to a streaming potential which is reported as a voltage. Titrant (0.1–1 mM poly(diallyldimethylammonium chloride) (polyDADMAC); Sigma Aldrich) is added to the sample until the point when all charges are neutralized and a zero streaming potential is reached. The specific charge density (*SCD*) was evaluated using the following equation:

$$SCD = -\frac{V_{\text{titrant}} \cdot S_{\text{titrant}} \cdot C}{m_{\text{particle}} \cdot SSA_{\text{particle}}}, \tag{1}$$

where $SCD$ is the specific charge density (C m$^{-2}$) of negative sign, $V_{titrant}$ is the volume of titrant (l) consumed to reach zero streaming potential, $S_{titrant}$ is the concentration of the titrant (M), $C$ is Faraday's constant (96485 C mol$^{-1}$), $m_{particle}$ is the mass of the particles (g), and $SSA_{particle}$ is the specific surface area of the particles (m$^2$ g$^{-1}$). The Brunauer–Emmett–Teller (BET) nitrogen adsorption method was used to determine the specific surface area of STx-1b as 83.8 m$^2$ g$^{-1}$.

*Supernatant solution composition.* After centrifugation, the elemental composition of the supernatant solution was analysed using Inductively Coupled Plasma Optical Emission Spectroscopy (ICP-OES, Thermo Scientific IRIS Intrepid II XDL, Thermo Electron Corporation, Waltham, MA, USA). The cations of interest were Y$^{3+}$ and Cu$^{2+}$ (externally added), as well as Ca$^{2+}$, Na$^+$, Mg$^{2+}$ which leach out from the montmorillonite surface. The instrument was regularly calibrated during the measurements using reference elemental standards. The equivalent of total cations sorbed or released were normalized with respect to the mass of mineral powder in each sample (eq g$^{-1}$), and calculated as:

$$Cation_{sorbed/released} = \frac{|c_{added} - c_{supernatant}| \cdot V_{sample} \cdot n}{m_{particle}}, \qquad (2)$$

where $c_{added}$ is the cation concentration added initially (M), $c_{supernatant}$ is the cation concentration in the supernatant (M), $V_{sample}$ is the volume of sample (l), $n$ is the valence factor (eq mol$^{-1}$) and $m_{particle}$ is the mass of the particles (g). Each set of triplicate samples was measured three times and we report the average values with one standard deviation. Adsorption isotherms, encompassing exchanged and adsorbed cations, are generated by plotting $Cation_{sorbed/released}$ versus $c_{added}$ which allows for easy comparison between cations of varying charges.

### 2.3.3 Barasym: Ion exchange and adsorption

The following procedure was applied to substitute NH$_4^+$, the charge-balancing cation in Barasym, by Ca$^{2+}$ (Rytwo et al., 1991). 1 wt % suspension of Barasym was prepared in 1 M CaCl$_2$ solution in a polypropylene tube. The particles exposed to the solution were allowed to equilibrate for 24 h on an agitation table. Following this, the suspension was centrifuged at 3000 rpm for 1 h, and the supernatant solution was discarded. Milli-Q water was added to the settled particles, agitated, centrifuged and the supernatant solution was again discarded. This process was repeated until the conductivity of the supernatant solution approached that of Milli-Q water. The settled particles were freeze dried in the same way as described in Sect. 2.3.1. The sample was again prepared in triplicates. A semi-quantitative evaluation of NH$_4^+$ exchange with Ca$^{2+}$ by Fourier Transform Infrared Spectroscopy (FTIR) showed that about 30 % of the NH$_4^+$ was replaced (See Appendix B).

### 2.4 Emulsion freezing experiments

### 2.4.1 Minerals freshly suspended in pure water

We described the general setup of immersion freezing experiments in our previous series of papers (Kumar et al., 2018; 2019a; 2019b). We repeat here the essential aspects for convenience. The freezing experiments were carried out using the differential scanning calorimeter (DSC, TA Instruments, Q10) setup (Zobrist et al., 2008). All original samples were suspended in water

 **Table 3.** Physicochemical properties of the investigated smectites

| Smectite | CEC (meq/100g) | Exchangeable cations | Octahedral charge | Tetrahedral charge | Unit layers (dry) | Unit layers (wet) | Lateral dimension |
|---|---|---|---|---|---|---|---|
| STx-1b | 84.4 [a] | $Ca^{2+}$ [a] | -0.68 [a] | 0.0 [a] | 7.7 [b] | 4.7 [c] | ~100 nm (50–700 nm) [d] |
| SAz-1 | 120 [a] | $Ca^{2+}$ [a] | -1.08 [a] | 0.0 [a] | 5.4 [b] | 4.3 [c]; 1–20 [e] | ~100 nm (50–1000 nm) [e] |
| SWa-1 | 80.6 [f] | $Ca^{2+}$, $Mg^{2+}$ | -0.18 [a] | -0.91 [a] | 5.8 [b] | 3.9 [c] | 500–2000 nm [g] |
| SHCa-1 | 43.9 [a] | $Ca^{2+}$ | -1.35 [a] | -0.22 [a] | 21.8 [b] | 6.9 [c] | – |
| SAu-1 | 69.0 [a] | $Mg^{2+}$, $Na^+$ [a] | -0.54 [a] | -0.38 [a] | – | – | 85 %: < 2000 nm [a]  75 %: < 200 nm [a] |
| Barasym | Ba: 70 [a]  NH4: 140 [a] | $NH_4^+$ | 0.01 [a] | -1.5 [a] | 19.7 [b] | 4.4 [c] | ~100 nm [b] |
| Laponite RD | 63 [f] | $Na^+$ [f] | -0.7 [f] | 0.0 [f] | – | 1 [f] | ~30 nm [f] |
| MX-80 | 85 [h] | $Na^+$ (62 %), $Ca^{2+}$ (26 %) [i] | -0.53 [j] | -0.02 [j] | 12.0 [k] | 5.0 [k] | ~100 nm (50–700 nm) [l] |

[a] Clay Minerals Society
[b] Stepkowska et al., 2004, average number of layers at 30°C and 50 % RH, water sorption (WS) from Table 2.
[c] Stepkowska et al., 2004, average number of layers at 30°C and 100 % RH, water retention (WR) from Table 2
[d] Veghte and Freedman, 2014
[e] Metz et al., 2005, AFM measurements, diluted and gently ultrasonicated sample, centrifuged onto crystal
[f] Delavernhe et al., 2018
[g] Stucki and Tessier, 1991
[h] Navarro et al., 2017
[i] Matusewicz et al., 2016
[j] values for SWy-1, SWy-2, and SWy-3 from CMS
[k] values taken for SWy-1 investigated in Stepkowska et al., 2004
[l] Matusewicz et al., 2019

(molecular biology reagent water from Sigma-Aldrich) and sonicated for 5 min before preparing the emulsions. The suspension concentrations are given in Table 4. The suspension and an oil/surfactant mixture (95 wt % mineral oil (Sigma Aldrich) and 5 wt % lanolin (Fluka Chemical)) were mixed at a ratio of 1:4 using a rotor-stator homogenizer (Polytron PT 1300D with a PT-DA 1307/2EC dispersing aggregate) for emulsification (40 s at 7000 rpm). A portion of this emulsion was placed in an aluminum pan, which was hermetically closed. As described by Marcolli et al. (2007), we performed three freezing cycles in the DSC. The first and the third freezing cycles were run at a cooling rate of 10 K min⁻¹ to control the stability of the emulsion. The second freezing cycle was run at a cooling rate of 1 K min⁻¹ and used for evaluation.

A typical DSC thermogram from freezing of an emulsion containing INPs exhibits two freezing signals (Kumar et al., 2018). The peak occurring at a higher temperature represents the heat release via heterogeneous freezing and the second peak occurring at a colder temperature is due to homogeneous freezing (Figure A). The freezing temperatures, $T_{het}$ and $T_{hom}$, are

determined as the onset of the heterogeneous and homogeneous freezing peak, respectively. Droplets as large as 12 μm in diameter are considered to be relevant for the heterogeneous freezing onset. The loading of these droplets with particles depends on the particle size distribution and the suspension concentration. $T_{hom}$ is precise within ±0.2 K. $F_{het}$ is defined as the ratio of the heterogeneous freezing signal to the total freezing signal (heterogeneous and homogeneous). More details about $T_{het}$, $F_{het}$ and the general evaluation procedure can be found in Kumar et al. (2018) and are also shown in Appendix A. The median droplet diameter in the emulsion is ≈2–3 μm in terms of number size distribution, but a broad distribution in terms of volume. Droplets with diameters of about 12 μm are considered to be relevant for the freezing onset in the DSC experiments (Marcolli et al., 2007; Kaufmann et al., 2016). The random spikes at temperatures warmer than the heterogeneous freezing onset are caused by the freezing of few large droplets (sometimes up to 300 μm present at the tail-end of the droplet size distribution) and are excluded from the evaluation. Their presence is likely due to the coalescence of some smaller droplets probably while transferring the sample to the aluminum pan for DSC and is not reproducible.

In the case of emulsions formed with pure water only, the droplets start to freeze at 237±0.2 K (Figure 1). However, when we introduce dust particles, the number of particles in a single droplet is governed by the volume of the droplet. With increasing droplet volume, the probability of accommodating particles in that droplet increases. Hence, the freezing of larger droplets dominates the heterogeneous freezing signal. While, the homogeneous freezing signal results either from the freezing of smaller empty water droplets or droplets containing particles which are ice nucleation inactive. $F_{het}$ reported in this study cannot be directly translated into absolute quantifiable parameters, rather it should be considered as a qualitative parameter to compare the efficacy of ice nucleation of different dust samples or to assess the changes in efficacy due to different treatments. We also use heterogeneous freezing onset to characterize the freezing temperature because it is a very well-defined parameter easily evaluated from the thermograms. A combination of $T_{het}$ and $F_{het}$ parameters provide a good measure of the overall ice nucleation abilities of the smectites.

In case of modified STx-1b, emulsion freezing experiments were performed using 1 wt % suspensions prepared from each of the triplicate set of samples. Mean and standard deviation for $T_{het}$ and $F_{het}$ obtained for the triplicate samples are reported. The modified samples were also compared with the original sample. In case of Barasym, we conducted freezing experiments using 0.5 wt % suspensions in pure water. Experiments were performed in triplicates for only few cases. Average precision in $T_{het}$ was ±0.2 K. The absolute uncertainties in $F_{het}$ are on average ±0.02 and do not exceed ±0.1. Furthermore, 0.5 wt % suspensions of modified Barasym were prepared with varying concentrations (0–0.5 wt %) of $(NH_4)_2SO_4$ (Sigma Aldrich, ≥ 99 %) and (0–0.05 molal) $NH_3$ solutions (Merck, 25 %) to assess the influence of externally added solute species on the IN ability of modified Barasym.

**2.4.2 Aging experiments**

To assess the effect of swelling on ice nucleation by montmorillonite, few modified STx-1b samples were aged for 2 to 3
320   months and for two years in the following two ways: aging in air and aging in water. Aging in air involved storing samples in
polypropylene tubes at room temperature, during which they interact with the water vapor in the tube, while aging in water
involved exposing the particles to pure water.

**3 Results**

**3.1. IN activity of the smectite samples**

325   Figure 1 compiles DSC thermograms from emulsion freezing experiments performed with the different smectite samples at
suspension concentrations between 1 and 5 wt %. Onset freezing temperatures ($T_{het}$) and heterogeneously frozen fractions
($F_{het}$) are given in Table 4. The peak with onset temperature at $237 \pm 0.2$ K marks homogeneous ice nucleation (as depicted by
the orange curve in Fig. 1), while signals at higher temperatures are due to heterogeneous freezing on INPs immersed in the
emulsion droplets. With the exception of the Laponite sample, all investigated smectites show a heterogeneous freezing signal,
330   yet with considerable variability in $T_{het}$ and $F_{het}$.

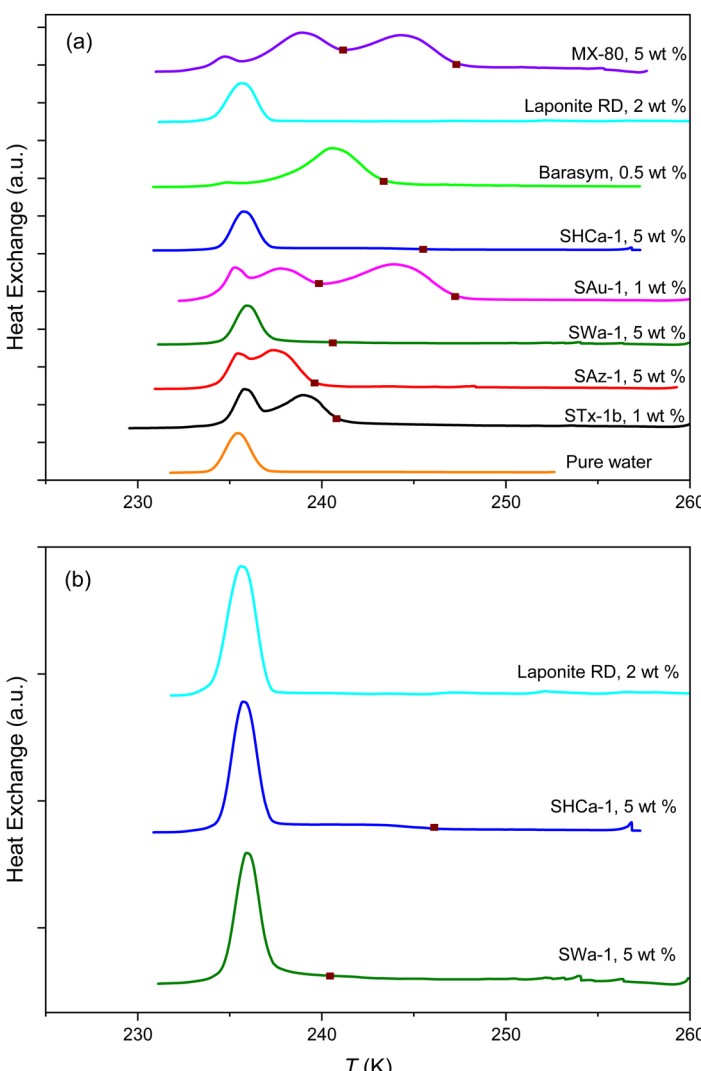

**Figure 1.** (a) DSC thermograms of all investigated smectites from emulsion freezing experiments at concentrations as indicated on the curves. For clarity, the highest freezing signals in all curves are normalized to the same height. For reference purposes a DSC thermogram for pure water (orange curve) is added, showing only homogeneous freezing signal. The brown squares depict the heterogeneous freezing onsets ($T_{het}$) for the samples. (b) DSC thermograms of three samples (Laponite RD, SHCa-1 and SWa-1) with either weak or no distinct heterogeneous freezing signals are shown separately for clarity.

The range of freezing temperature observed for various smectites are in agreement with previous studies carried out with similar dust loadings of water droplets (Welti et al., 2009; Atkinson et al., 2013). All IN active smectites with the exception of Barasym and the hectorite SHCa-1 exhibit a freezing peak with $T_{het}$ around 240 K (exact range of 239.9–241.2 K, see Table

4). A freezing peak with such an onset has also been observed in emulsion freezing experiments with the acid treated K10 and KSF montmorillonites from Sigma Aldrich and the Wyoming montmorillonite (SWy-2). This freezing peak also coincides with the one found for kaolinites KGa-1b and KGa-2 (both from CMS) suspensions, and was referred to as the standard freezing peak by Pinti et al. (2012). Two of the smectites with an onset freezing temperature at around 240 K exhibit a second freezing peak at higher temperatures, namely the bentonites SAu-1 and MX-80 with the higher $T_{het}$ at 247.1 K and 247.3 K, respectively. Pinti et al. (2012) observed a similar second freezing peak for SWy-2 at 247.0 ± 0.2 K in emulsion freezing experiments and referred to it as special peak. Atkinson et al. (2013) hypothesized that this special freezing peak in SWy-2 may not be due to montmorillonite but arise from the feldspar impurity (16 w t%) within the sample. Despite having the same source as SWy-2, MX-80 contains only a minor feldspar impurity of 3.1 wt %. The bentonite SAu-1 exhibits a feldspar impurity of 3.5 wt %. These feldspar fractions seem too little to explain the strong special peaks. Both SAu-1 and MX-80 contain quartz as the most abundant impurity with 6.7 wt % in SAu-1 and 4.8 wt % in MX-80. Quartz proved to be highly IN active with freezing onsets at about 247–250 K and a heterogeneously frozen fraction of $F_{het}$ = 0.79 in emulsion freezing experiments with 1 wt % suspensions, when the sample was freshly milled. Yet, the IN activity vanished with time when the quartz sample was aged in water (Kumar et al., 2019a). Therefore, the IN activity of quartz has been ascribed to defects (Kumar et al., 2019a, Zolles et al., 2015), implying that the quartz impurities within the bentonites need to be highly defective to account for such strong freezing signals. On the other hand, the clay mineral illite proved to be IN active up to $T_{het}$ = 244.7 ± 0.1 K in case of illite NX and even up to $T_{het}$ = 249.9 K for illite SE. Illite NX and SE consist of 86 wt % and 77 wt % illite, respectively, with kaolinite as the major impurity (10 wt %) and only traces of feldspars and quartz. Moreover, the kaolin mineral halloysite exhibited IN activity up to 245 K in emulsion freezing experiments for samples without quartz or feldspar impurities (Klumpp et al., 2023). Thus, the special peaks of SAu-1 and MX-80 are within the freezing range observed for clay minerals in emulsion freezing experiments, and an origin from smectite INPs is likely.

**Table 4.** Ice nucleation ability of investigated smectites

| Smectite | Concentration (wt %) | $T_{het}$ (K)[a] | $F_{het}$[a] |
|---|---|---|---|
| | 1.0 | 240.3 | 0.61 |
| STx-1b | 2.0 | 241.2 | 0.78 |
| | 5.0 | 240.7 | 0.87 |
| SAz-1 | 5.0 | 239.9 | 0.72 |
| SWa-1 | 5.0 | 240.4 | 0.17 |
| SAu-1 | 1.0 | 247.1 / 239.9 | 0.84 |
| SHCa-1 | 5.0 | 246.6 | 0.16 |
| Barasym | 0.5 | 243.2 | 0.95 |
| Laponite RD | 2.0 | – | – |
| MX-80 | 5.0 | 247.3 / 241.1 | 0.91 |

[a]uncertainty does not exceed 0.5 K and 0.1 for $T_{het}$ and $F_{het}$, respectively


Barasym, a synthetic montmorillonite, shows the highest heterogeneously frozen fraction with $F_{het} = 0.95$ and $T_{het} = 243.2$ K for the 0.5 wt % suspension. This high IN activity can be traced back to its exchangeable cation $NH_4^+$, since low concentrations of $NH_4^+$ or $NH_3$ have already been shown to enhance the IN activity of kaolinite (Kumar et al., 2019b). While this points to a specific role of the cations, comparison of the IN activity of the other smectites questions such a relationship again. STx-1b,

SAz-1, SWa-1, and SHCa-1 are all purely $Ca^{2+}$ or rich in $Ca^{2+}$, yet, they show strong variability in both $T_{het}$ and $F_{het}$. When we compare the IN activity of Laponite RD and MX-80, both rich in $Na^+$, we find that the Laponite is completely inactive while MX-80 shows a very high IN activity. Yet, as these samples also differ in other aspects such as charge distribution between layers and total CEC depending on the degree and location of isomorphic substitution, the effect of the exchangeable cation could be concealed. We therefore decided to investigate the role of cations more specifically by performing ion exchange

experiments as described in the following sections.

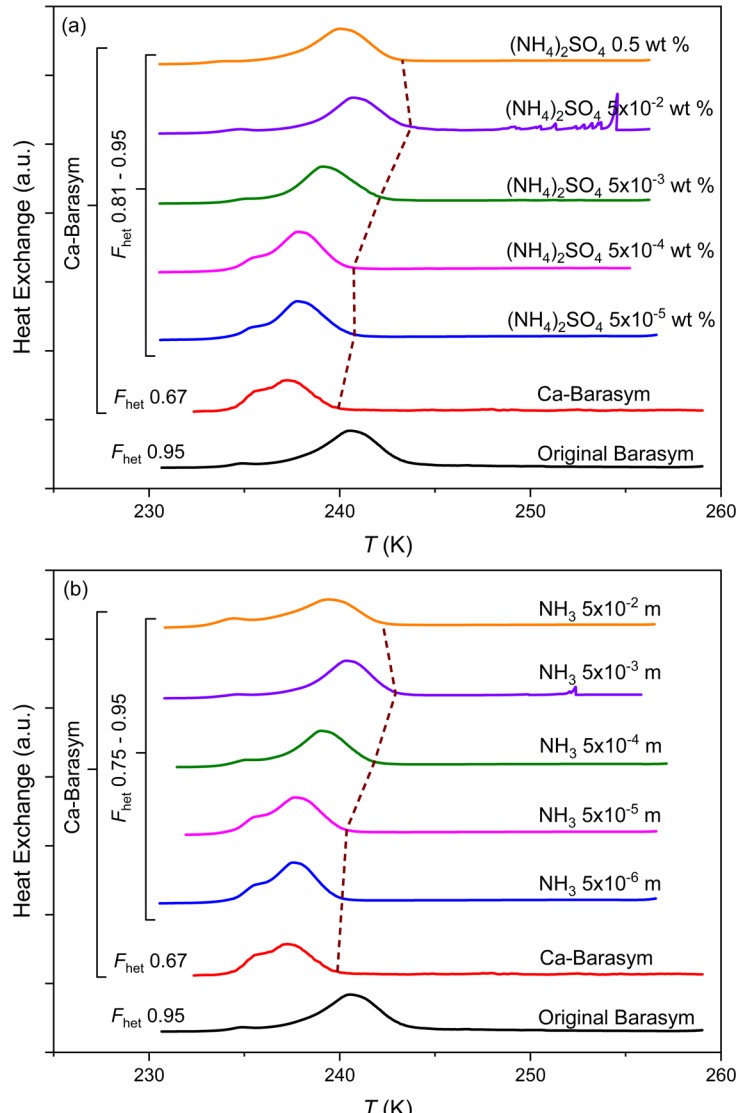

**Figure 2.** DSC thermograms obtained via emulsion freezing experiments with Barasym in pure water (original Barasym), compared with the ion exchanged Ca-Barasym in pure water and in (a) ammonium sulfate solutions (in wt %) and (b) ammonia solutions (in molality) with increasing concentration as indicated on the DSC curves. The concentration of Barasym in the suspensions was 0.5 wt %. The dashed brown line indicates the freezing onsets. All curves are normalized with respect to the total areas under the heterogeneous and homogeneous freezing curves.

## 3.2 Ion exchange experiments with Barasym

To investigate how ammonium as the charge balancing cation impacts the IN activity of Barasym, we replaced it by $Ca^{2+}$ as described in Sect. 2.3.3. As can be seen in Fig. 2, this substitution goes along with a decrease in IN activity. A 0.5 wt % suspension of Ca-Barasym in pure water exhibits $T_{het} = 239.5$ K and $F_{het} = 0.67$ compared with $T_{het} = 243.2$ K and $F_{het} = 0.95$ for the original Barasym at the same concentration. Thus, after ion exchange, the IN activity of Barasym becomes similar to the ones of the montmorillonites STx-1b and SAz-1. The IR spectra (See Appendix B) shows evidence that only around 30 %

of ammonium has been replaced, but, as it seems, this exchanged fraction is responsible for the enhancement in IN activity observed for Barasym. In addition, there is clear evidence from previous studies that smectites have strong affinity for ammonium (Nommik and Vahtras, 1982; Dontsova et al., 2005; Alshameri et al., 2018). Adding again small amounts of either $(NH_4)_2SO_4$ or $NH_3$ to the Ca-Barasym suspension brings $T_{het}$ and $F_{het}$ back to the values of the original Barasym for solute concentrations around $5 \cdot 10^{-3}$ molal $NH_3$ or $5 \cdot 10^{-2}$ wt % $(NH_4)_2SO_4$. Improved IN ability of montmorillonite (K10 from Sigma-

Aldrich) has also been reported by Salam et al (2007; 2008) in freezing experiments with a continuous flow diffusion chamber for particles that previously had been exposed to $NH_3$ gas. This enhancement seems to be a general feature of the ice nucleation sites of aluminosilicates as it has also been observed for kaolinites, halloysites, micas, and the feldspars microcline, sanidine, and andesine (Kumar et al., 2018; 2019b; Klumpp et al., 2023), but not for quartz or biogenic INPs such as bacteria, fungi or humic substances (Worthy et al., 2021). This points to similarities in the chemical makeup of nucleation sites of

aluminosilicates in general.

## 3.3 Ion exchange experiments with STx-1b

To gain further insight into the role of the charge-balancing cations for the ability of smectites to nucleate ice, we exchanged the natural cations of the montmorillonite STx-1b once with $Y^{3+}$ and once with $Cu^{2+}$ and re-assessed the IN activity of the ion-exchanged samples. Ion exchange was performed stepwise and monitored with ICP-MS and through measuring the specific

charge density on the particle surface. Figure 3 shows the results for ion exchange with $Y^{3+}$. Based on the CEC reported by CMS, full ion exchange is reached at a concentration of around 1.0 mmolal $Y^{3+}$ (Fig. 3(a)). As the trivalent $Y^{3+}$ has a higher affinity for the montmorillonite surface than the bivalent $Ca^{2+}$, we expect full exchange of $Y^{3+}$ with $Ca^{2+}$. We added the $Y^{3+}$ solution in small portions up to this value and completed the data series by two points where $Y^{3+}$ is present in excess. The ICP-MS measurements show that $Ca^{2+}$ as the major charge balancing cation is released together with minor amounts of $Mg^{2+}$ and

$Na^+$ while $Y^{3+}$ is sorbed. As the sum of ions released exceeds the amount of sorbed $Y^{3+}$, the surface charge, which was -0.03 $C\ m^{-2}$ for the original STx-1b becomes less negative and approaches zero with increasing $Y^{3+}$ concentration. Adding $Y^{3+}$ in excess leads to more adsorption of $Y^{3+}$ but does not seem to release more cations, thus confirming that ion exchange is complete at the indicated CEC. The DSC thermograms measured for each sample in triplicates at 1 wt % suspension concentration reveals quite a large scatter in $T_{het}$ and $F_{het}$ between the samples but no clear trend towards higher or towards lower IN activity

with increasing ion exchange. Yet, the ion-exchanged samples exhibit higher $F_{het}$ and $T_{het}$ with average values of $0.68 \pm 0.10$

and $240.2 \pm 0.2$ K, respectively, compared with $F_{het} = 0.61 \pm 0.06$ and $T_{het} = 240.3 \pm 0.5$ K for the STx-1b sample taken directly out of the box. This also applies to the blank sample which was treated the same way as the ion-exchanged samples including agitation for 72 h in suspension, centrifugation and lyophilization, for which $F_{het} = 0.77 \pm 0.03$ and $T_{het} = 240.3 \pm 0.2$ K was measured. Yet, when we repeated the DSC measurements after having aged the ion-exchanged samples for 2–3 months in

water, the IN activity of almost all samples decreased below the value of the original sample, with average $F_{het} = 0.41 \pm 0.13$ and $T_{het} = 239.3 \pm 0.4$ K, again including the zero-exchange sample and with no clear dependence on the degree of ion exchange.

Figure 4 shows the same type of experiments for ion exchange with $Cu^{2+}$ instead of $Y^{3+}$. Based on the CEC, we expect complete ion exchange at about 1.5 mmolal $Cu^{2+}$, while the sorption curve of $Cu^{2+}$ plateaus at a slightly higher value of (1.67 mmolal

$Cu^{2+}$). This difference is most probably due to surface adsorption as made evident by the decrease in negative surface charge. Yet, also the replacement of the natural charge-balancing ions with $Cu^{2+}$ does not seem to influence the IN activity, which reaches a constantly high value above the one of the original STx-1b, namely $F_{het} = 0.74 \pm 0.05$ and $T_{het} = 240.5 \pm 0.3$ K. We tested the IN activity of the samples again after aging for 2–3 months in water, yet, for $Cu^{2+}$ as the exchangeable cation, there was no decrease in IN activity with time in water when the $Cu^{2+}$ concentration was above 0.6 mmolal and only a slight decrease

for lower concentrations.

To further explore the effect of aging in water, we repeated the emulsion freezing experiments with original STx-1b exposed to liquid water for 24 h, 48 h, 72 h, 4 months, 7 months and 2 years. With some variability, aging in water hampered the IN ability as observed through reduced $F_{het}$ for samples aged up to 7 months (shown in Fig. 5). However, the heterogeneous freezing peak was still clearly observed. In contrast, the samples aged in water for 2 years yielded a completely different DSC

thermogram (shown in Fig. 5) with an overall lower $F_{het} \approx 0.33$ and a small freezing peak with onset $T_{het} \approx 246$ K, while the freezing peak at 240.3 K is reduced to a weak tail of the homogeneous freezing peak, which starts at ~239 K.

Overall, the ion-exchange measurements show that the charge-balancing ion does not have a direct impact on IN activity. Yet, there seems to be an indirect influence as the IN activity of samples after 2–3 months in water is decreased for $Ca^{2+}$ and $Y^{3+}$ as the exchangeable cations, but not for $Cu^{2+}$. To understand this behavior, we investigate the relationship between IN activity

and particle morphology in the following section.

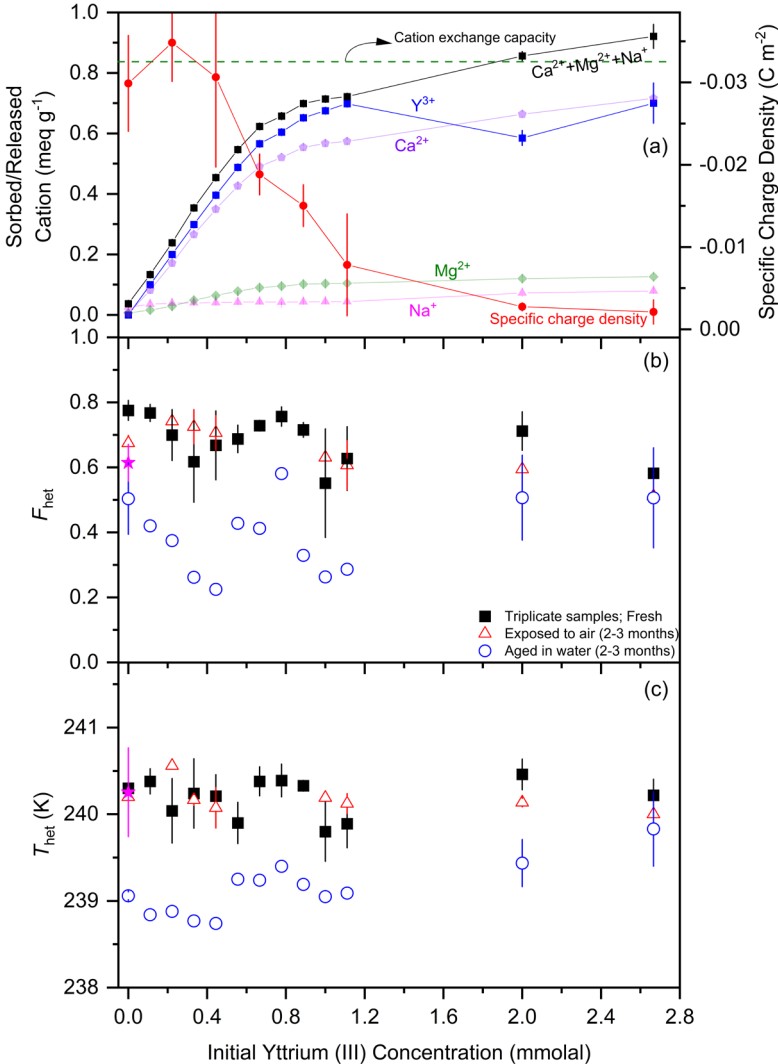

**Figure 3.** Ion exchange experiments as a function of $Y^{3+}$ concentration in a 0.5 wt % STx-1b suspension showing the effect of the stepwise exchange of the charge-balancing cations ($Ca^{2+}$ and minor amount of $Mg^{2+}$ and $Na^+$) with $Y^{3+}$ on the IN activity of the montmorillonite. Panel (a) shows sorbed $Y^{3+}$ together with released $Ca^{2+}$, $Mg^{2+}$ and $Na^+$ as separate curves and sum to $Ca^{2+} + Mg^{2+} + Na^+$ in terms of meq/g (left scale) together with the specific charge density on the montmorillonite particle surface (right scale). The horizontal green dashed line marks the CEC as reported by CMS. DSC thermograms of the freshly ion-exchanged and lyophilized samples as 1 wt % suspensions in pure water have been measured in triplicates (black filled squares with error bars representing ± 1 SD) a few days after preparation and been evaluated in terms of $F_{het}$ (panel (b)) and $T_{het}$ (panel (c)). A few samples have been re-measured after 2–3 months either after storage in air (red open triangles) or aged in water (blue open circles). The black square at 0 mmolal $Y^{3+}$ marks the sample that was treated the same way as the ion-exchanged samples (including lyophilisation). As a reference, filled magenta stars indicate $F_{het}$ and $T_{het}$ of the original STx-1b in water.

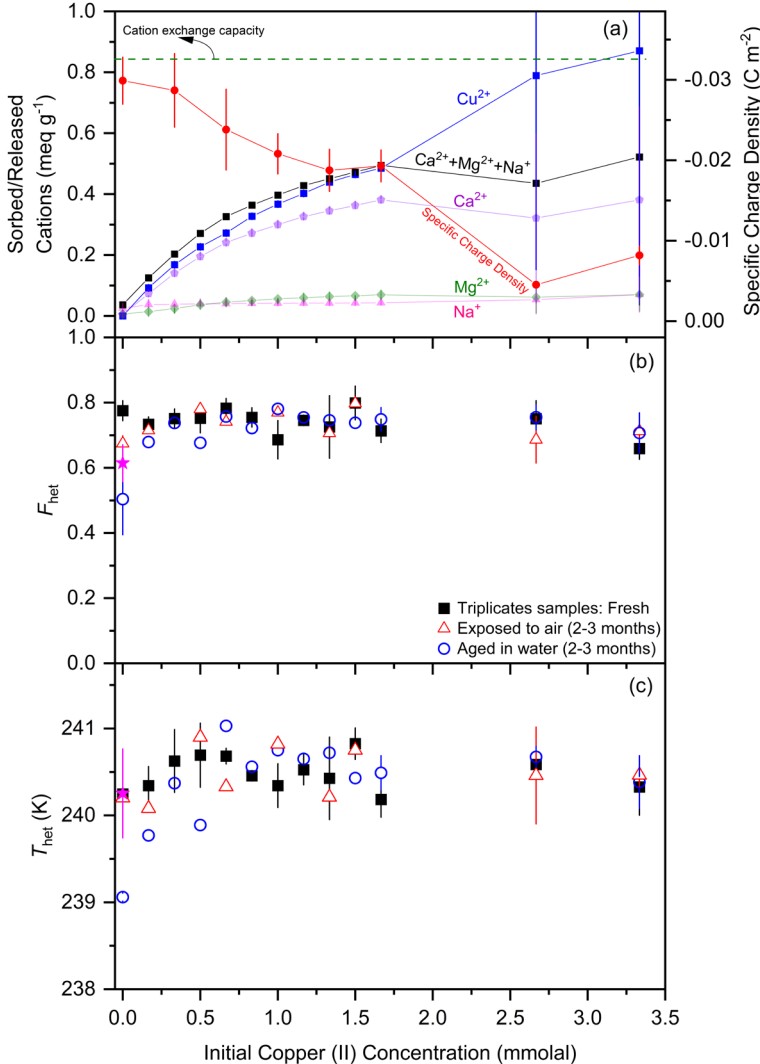

**Figure 4.** Ion exchange experiments as a function of $Cu^{2+}$ concentration in a 0.5 wt % STx-1b suspension showing the effect of the stepwise exchange of the charge-balancing cations ($Ca^{2+}$ and minor amount of $Mg^{2+}$ and $Na^+$) with $Cu^{2+}$ on the IN activity of the montmorillonite. Panel (a) shows sorbed $Cu^{2+}$ together with released $Ca^{2+}$, $Mg^{2+}$ and $Na^+$ as separate curves and sum to $Ca^{2+} + Mg^{2+} + Na^+$ in terms of meq/g (left scale) together with the specific charge density on the montmorillonite particle surface (right scale). The horizontal green dashed line marks the CEC as reported by CMS. DSC thermograms of the freshly ion-exchanged and lyophilized samples as 1 wt % suspensions in pure water have been measured in triplicates (black filled squares with error bars representing ± 1 SD) a few days after preparation and been evaluated in terms of $F_{het}$ in panel (b) and $T_{het}$ in panel (c). A few samples have been re-measured after 2–3 months either after storage in air (red open triangles) or aged in water (blue open circles). The black square at 0 mmolal $Y^{3+}$ marks the sample that was treated the same way as the ion-exchanged samples (including lyophilisation). As a reference, filled magenta stars indicate $F_{het}$ and $T_{het}$ of the original STx-1b in water.

# 4 Discussion

## 4.1 Exchangeable cations and ion adsorption

As smectite surfaces are highly populated with exchangeable cations, one could expect that they impact the IN ability of these surfaces since ion charge and size should influence the way a surface interacts with water molecules. Therefore, we chose our smectite samples to cover different exchangeable cations. $Ca^{2+}$ is the prevailing charge-balancing cation in most of the investigated smectites, namely in the montmorillonites STx-1b, and SAz-1, the nontronite SWa-1, and the hectorite SHCa-1. Conversely, the bentonite MX-80 is rich in $Na^+$ with a minor contribution of $Ca^{2+}$, while the synthetic Laponite RD is a pure

$Na^+$ smectite. The bentonite SAu-1 is rich in $Mg^{2+}$ with a minor contribution of $Na^+$. Finally, the synthetic mica-montmorillonite Barasym carries $NH_4^+$ as the exchangeable cation. Yet, there is no recognizable correlation between IN activity and exchangeable cation. While the Na-rich bentonite MX-80 exhibits a high IN activity, the pure $Na^+$ Laponite proved to be IN inactive. There is also a large variability between the IN activity of the Ca-rich smectites with SWa-1and SHCa-1 being at the low end in terms of heterogeneously frozen fraction ($F_{het}$ of 0.17 and 0.16, respectively, for 5 wt % suspensions in pure

water) and STx-1b ($F_{het} = 0.61$ for a 1 wt % suspension) and SAz-1 ($F_{het} = 0.72$ for a 5 wt % suspension) at the high end. As the samples also differ in terms of mineralogical impurities and magnitude and location of layer charge, we performed ion-exchange experiments with Barasym and STx-1b to exclude such differences and to isolate the effect of the charge-balancing cations. While, the exchange of $NH_4^+$ by $Ca^{2+}$ in Barasym revealed a clear enhancing effect of $NH_4^+$ on the IN activity of the smectite as discussed in Sect. 3.2, the exchange of the charge-balancing cations of STx-1b either with $Y^{3+}$ or $Cu^{2+}$ showed no

clear effect on the IN activity of the freshly prepared samples. This is opposite to the high sensitivity to the presence of low concentration of alkali cations that was found for feldspars (Yun et al., 2020; Kumar et al., 2018; 2019b), and also against expectations if one considers how highly populated the smectite surfaces are with exchangeable cations. To resolve this conundrum, we take a closer look at the impact of the exchangeable cations on smectite surface properties in the forthcoming sections.

It has been previously established that feldspars can undergo cation exchange and adsorption when exposed to salt solutions (Nash and Marshall, 1957; Demir et al., 2001; 2003). Contrary to laboratory experiments, classical molecular dynamics (MD) simulations have not been able to capture ice nucleation on K-feldspar surfaces. The (001), (010) and (100) surfaces of K-feldspar exposed to various salt solutions (including ammonium-containing solutions) show that sorbed ions affect interfacial water orientation, albeit unfavorable for ice nucleation (Kumar et al., 2021). MD simulations performed on the basal Al surface

of kaolinite exposed to various solutions revealed that ions near the surface reduce the amount of clear surface area available to serve as an ice template (Ren et al., 2020). On the other hand, our study shows that within measurement uncertainty, the IN ability of fresh STx-1b is not affected when cation concentrations are close to or even surpass the CEC limit where surface and interfacial ion concentrations might be high.

## 4.2 The role of the basal surfaces

Since smectites are present as thin platelets, the basal surfaces dominate the total surface area. One type of basal surfaces consists of Si–O–Si bridges forming hexagonal rings (Kumar et al., 2019; Bear, 1965). In case of uncharged clay minerals, such as kaolinite, these siloxane surfaces are hydrophobic with contact angles between 100° and 110° (Šolc et al., 2011; Szczerba et al., 2020). Yet, as smectites carry negative charge either in the octahedral sheet (montmorillonites), the tetrahedral sheet (beidellites and hectorites) or in both (montmorillonites or beidellites depending on the ratio), the interlayer space and

the external basal surfaces are densely populated with exchangeable cations. Using molecular dynamics simulations, Szczerba et al. (2020) showed that the hydrophilic nature of smectite surfaces stems from the hydration of the adsorbed cations rather than from the negative layer charge. They found that a charged montmorillonite surface with the charge in the octahedral sheet was only slightly more hydrophilic than the uncharged one (reduction in contact angle by about 15°). If the negative charge resided in the tetrahedral sheet, the effect was larger (contact angle of 60°). Yet, when the siloxane surface was populated by

counterions to balance the negative charge, the surface became highly hydrophilic. Zheng and Zaoui (2017) determined average contact angles of 25° when they deposited water nanodroplets on top of Na-montmorillonite in molecular dynamics simulations. These findings agree with molecular dynamic simulations by Yi et al. (2018), who showed that the montmorillonite surface has a very weak adsorption energy for water, but a very strong electrostatic attraction for exchangeable cations, which possess a hydration shell of several water layers with exact dimensions depending on the charge and size of the

adsorbed cations. In the case of Na-montmorillonite, the hydration shell was modelled to consist of about 0.85 nm thick ordered layer corresponding to three layers of water molecules followed by a 0.89 nm transition layer, and only at a distance >1.74 nm water molecules exhibited bulk properties. The bound water in the ordered layer of smectites has higher density and viscosity, and decreased mobility and self-diffusion compared with bulk water (Ricci et al., 2015; Wang et al., 2020; Zhang and Pei, 2021). Such densely bound water layers do not seem to provide a medium that is suited for ice nucleation. Lata et al. (2020)

investigated the ability of the siloxane surface of muscovite mica to direct water molecules into ice-like configurations. Like smectites, muscovite possesses a negative layer charge that, in natural muscovite, is balanced by $K^+$. As these cations hold the muscovite layers together, this mineral does not swell and only the cations on the external siloxane surface are exchangeable. Molecular dynamics simulations indicated that ice-like clusters preferentially formed in surface regions that are devoid of cations but these were unable to grow to critical size. Thus, the cations do not seem to endow the basal surfaces with IN

activity, rather, they hinder it.

## 4.3 The role of the edges

Edges constitute only a minor part of the smectite surfaces. Yet, as they carry reactive hydroxy groups with pH dependent charge, they play a key role in processes occurring on smectites, such as sorption of inorganic ions and organic species, and dissolution of smectites (Sanders et al., 2010; Segad et al., 2010; 2012; Newton et al. 2015).

As the Si-tetrahedral sheets are terminated by silanol (Si-OH) functional groups, and the dioctahedral voids of the Al-octahedral sheets are filled with hydroxy groups, hydroxylated chains form at the edges such as (see Fig. 5 of White and Zelazny (1988)):

- Si(OH)–O–Al(OH$_2$)$_2$$^+$–O–Si–OH at acidic conditions (pH ~ 3 – 4.5),

- Si(OH)–O–Al(OH)$_2$$^-$–O–Si–OH at slightly acidic to neutral conditions (pH ~ 6.5), and

- Si(OH)–O–Al(O)$_2$$^{3-}$–O–Si–OH at slightly basic conditions (pH ~ 7.5)

The point of zero charge of edge sites is close to pH 6. As the edges are highly hydroxylated, they can form hydrogen bonds with water molecules and therewith fulfill a key prerequisite for ice nucleation (Pedevilla et al., 2017). Moreover, the hydroxylated edges are the only common surface of the clay minerals illite, kaolinite, and smectite, which all exhibit IN activity in a similar temperature range (Pinti et al., 2012), and also respond with an enhancement in IN activity to the presence of

ammonium and ammonia (Kumar et al., 2019b; Klumpp et al., 2023). Thus, the hydroxylated edges are the most likely location for ice nucleation, as also has been proposed by Klumpp et al. (2023), who analyzed the IN activity of kaolinites and different halloysite nanotubes. They found that the hydroxylated edges need to be broad enough to host a critical ice embryo and that the wall thickness of the halloysite tubes is a key parameter for the freezing temperature. With a thickness of ~0.96 nm (Nicola et al., 2021), a single T—O—T layer is too thin to host a critical ice embryo, based on estimates from CNT, which predicts

critical sizes of around 15–20 nm$^2$ assuming a spherical surface area (Kaufmann et al., 2017; Qiu et al., 2019). Thus, several smectite layers need to be stacked together to reach the required size for ice nucleation. In fully hydrated smectites, 2–4 water layers are in-between the smectite layers, leading to a periodic spacing of about 1.5–2 nm (Segad et al., 2012). Thus, to provide a sufficiently large area for ice nucleation, at least three smectite layers need to be stacked together. Moreover, these layers should be properly aligned to form a coherent hydroxylated surface. As smectites are turbostratic, with adjacent layers that are

randomly rotated or translated with respect to one another (Cases et al., 1992; Segad et al., 2010), particle edges are often rugged and irregular (Stepkowska et al., 2004) such that the condition of a coherent hydroxylated surface is not necessarily fulfilled.

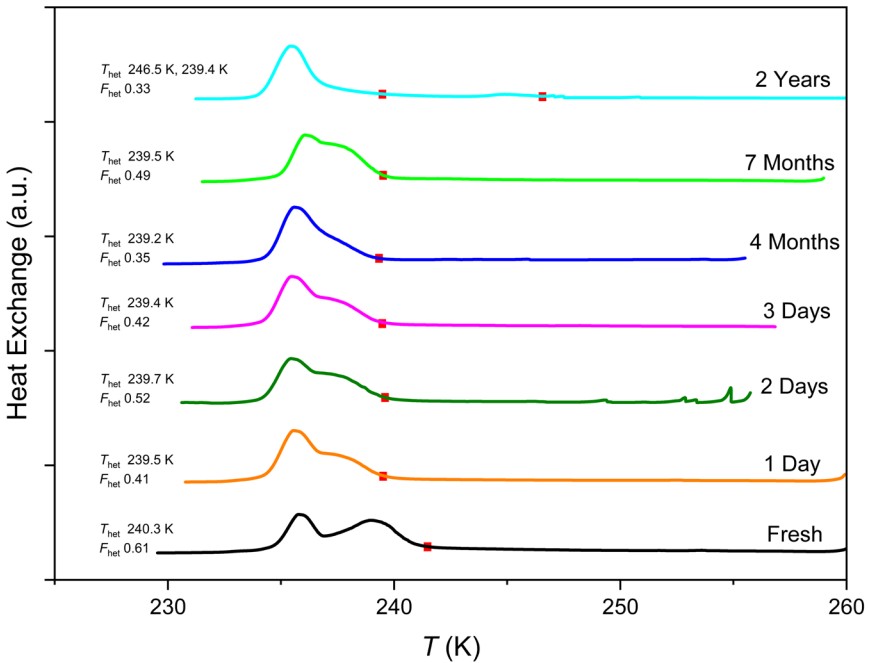

$T_{het}$ 246.5 K, 239.4 K
$F_{het}$ 0.33

$T_{het}$ 239.5 K
$F_{het}$ 0.49

$T_{het}$ 239.2 K
$F_{het}$ 0.35

$T_{het}$ 239.4 K
$F_{het}$ 0.42

$T_{het}$ 239.7 K
$F_{het}$ 0.52

$T_{het}$ 239.5 K
$F_{het}$ 0.41

$T_{het}$ 240.3 K
$F_{het}$ 0.61

2 Years

7 Months

4 Months

3 Days

2 Days

1 Day

Fresh

Heat Exchange (a.u.)

$T$ (K)

**Figure 5.** DSC thermograms of original STx-1b suspended in water and measured directly after preparation (fresh) and on the displayed days. Filled red squares mark the heterogeneous freezing onset temperatures ($T_{het}$). All curves are normalized with respect to the total areas under the heterogeneous and homogeneous freezing curves.

## 4.4 Swelling and delamination

When dried smectites are exposed to humid conditions, they take up water and swell. The dominant driving force for swelling is the hydration of the cations in the interlamellar space between smectite layers (Cases et al., 1992; Peng et al., 2019). Consequently, the valency and size of the exchangeable cations govern the RH dependence and degree of swelling together with the location and the value of the layer charge (Cases et al., 1992; Bérend et al., 1995). Smectites dried at room temperature keep some water bound to the exchangeable cations that is only removed upon heating (e.g. for Na-montmorillonite it is ~1

water molecule per Na$^+$). Increasing RH results in further hydration of the cations on the external surfaces and in the interlamellar space of tactoids leading to the expansion of the platelets. This process is referred to as crystalline swelling and occurs in discrete steps and is cation specific (Kahr et al., 1990; Jacinto et al., 2012; Zhang et al., 2014; Woodruff and Revil, 2011). Depending on the exchangeable cation, water uptake may be enhanced by osmotic swelling at high RH (e.g. for Na$^+$) to up to four water layers in-between the clay layers. As one water layer is about 0.3 nm thick, the dry spacing of smectite

tactoids of about 1 nm per layer increases up to 2.2 nm (Rao et al., 2013, Metz et al., 2005). Moreover, hydration of the interlamellar cations may also lead to the splitting of thicker tactoids (>20 layers) into thinner ones (<10 layers). The degree

of splitting depends on the exchangeable cation with $Na^+$-substituted smectites being split on average in thinner tactoids than $Ca^{2+}$-substituted ones (Mystkowski et al., 2000). In aqueous suspensions of high ionic strength, only crystalline swelling occurs, while in pure water or dilute solutions osmotic swelling may lead to full delamination of tactoids into single layers resulting in colloidal suspensions and gels (Metz et al., 2005; Segad et al., 2012). Besides the exchangeable cation, also the ionic strength of the solution, the surface charge and the basal dimensions of the platelets influence the degree and the timescale of delamination, which may take months, but in case of $Na^+$-smectites consisting of platelets of small lateral dimensions suspended in water may occur within minutes (Lagaly, 2006; Assemi et al., 2015; Delavernhe et al., 2018; Suman and Joshi, 2018; Pujala and Bohidar, 2019). When water is removed from smectite tactoids that have been dispersed into individual layers during infinite osmotic swelling, they rebuild again, and, depending on the drying method, form even larger aggregates than before (Metz et al., 2005). Apart from the drying method, the thickness of the platelets can also be altered through sonication (de Carvalho et al., 1996; Pacula et al., 2006).

## 4.5 Connection between particle thickness and IN activity

Given the partial or even complete delamination of smectites in pure water, the presence of a large enough edge surface to host the critical ice embryo needs to be considered as a limiting factor for the IN activity of smectites. Mystkowski et al. (2000) found a lognormal distribution of crystallite thickness for a series of smectites including nontronites and a hectorite sample through analysis of XRD reflection. For ethylene glycol saturated samples, which should represent air-dried and dehydrated smectites, they obtained mean thicknesses ranging from 3.4–7.3 layers. TEM analysis indicated that the mean thickness decreases at higher humidity and, with $Na^+$ as exchangeable cation, there was a high content of monolayers (30–49 % of counts) present (Mystkowski et al., 2000). Table 3 shows the average number of layers within tactoids of most of the smectites investigated in this study at 30 and 100 % RH as reported by Stepkowska et al. (2004). Yet, the whole distribution of tactoid thickness needs to be considered to judge the IN activity of a sample. We assume that at least three platelets need to be stacked within a tactoid to be IN active, and, with an increasing number of layers, the ice-nucleation temperature increases. These assumptions are in agreement with the estimates of the critical area of a nucleation site based on classical nucleation theory and molecular simulations (Kaufmann et al. 2017; Qiu et al., 2019). Applying these assumptions to the investigated smectites leads to the following connections between particle morphology and IN activity:

i.  The synthetic Laponite has been described to delaminate fully within few minutes when it is suspended in deionized water (Nicolai and Cocard, 2000). Thus, the absence of IN activity in Laponite suspensions can be easily explained by the too thin edges of single Laponite layers (1 nm).

ii.  The bentonite MX-80 is also rich in $Na^+$ and should therefore delaminate completely. Yet, a high IN activity with $T_{het}$ = 247.3 K / 241.1 K, and $F_{het}$ = 0.91 was measured for 5 wt % suspensions. Since no platelet thickness information could be found for this sample in literature, we use investigations on SWy montmorillonite provided from CMS to derive layer thickness. This seems justified as MX-80 and SWy-2 are sourced from the same mine and show similar freezing patterns in emulsion freezing experiments, both featuring two freezing peaks with onsets at 247.1 K and

241.1 K for MX-80 and 247.0 ± 0.2 and 238.2 ± 0.4 K for SWy-2 (Pinti et al., 2012). While Stepkowska et al. (2004) just determined the average thickness of SWy-1 tactoids, which yielded 12 unit layers in dry and 5 unit layers in wet conditions (see Table 3), size-resolved measurements by Assemi et al. (2015) revealed a bimodal distribution of SWy-2 platelets with mean equivalent spherical diameters of ~60 and 250 nm. They found variation in particle thickness and delamination behaviour, which they attributed to size-dependent elemental composition and surface charge of the platelets. The derived average tactoid thickness of the smaller particle population equalled 2.6 nm, pointing to fully delaminated particles that prevail in this size fraction. Conversely, the larger particle population exhibited an average tactoid thickness of ~36 nm. Such thick platelets would provide the required edge surface area for freezing occurring at ~247 K as has been observed for MX-80 and SWy-2.

iii.    For SHCa-1, the hectorite sample with a large share of calcite, $T_{het}$ = 246.6 K, and $F_{het}$ = 0.16 was measured for a 5 wt % suspension. The particles of this hectorite are among the thickest, yet with irregular rugged edges, which might be due to cementation by calcite (Stepkowska et al. (2004). Rugged edges together with cementation by calcite may explain both, the low heterogeneously frozen fraction as the edges might only form few hydroxylated surface areas that are sufficiently large to host an ice embryo, but some of these areas might be quite large, which again would explain the relatively high $T_{het}$.

iv.    SAz-1, on the other hand, exhibits a rather low onset freezing temperature ($T_{het}$ = 239.9 K) together with a rather large heterogeneously frozen fraction ($F_{het}$ = 0.72 for 5 wt % suspensions). Stepkowska et al. (2004) report thin particles at 30 % RH (5.4 unit layers on average) that only show limited delamination at 100 % RH to 4.3 unit layers on average. A detailed description of the morphology of SAz-1 has been presented in Metz et al. (2005), who measured the heights at different positions of single particles with AFM. They found that most of the particles consisted only of a single layer (25 out of 37). Further, they discriminated between single-level and multi-level particles, and found that single-level particles ranged from 1 to 6 layers, and multilayer particles were found to range from 3–6 to 17–20 layers. Thus, the edge surface area provided by these particles again seems to limit the IN efficiency.

v.    SAu-1 shows two freezing peaks with onsets at $T_{het}$ = 247.1 and 239.9 K. It carries $Mg^{2+}$ as major exchangeable cation and has been attributed a low swelling capability (Churchman et al., 2002), which can explain its high $F_{het}$ of 0.77 for 1 wt % suspensions. As it is interstratified with non-swelling illite layers (about 10 %), these may account for the freezing peak at 247.1 K, which is in the range observed for illite samples in emulsion freezing experiments (Pinti et al., 2012).

vi.    With $F_{het}$ = 0.17 and $T_{het}$ = 240.4 K, SWa-1 exhibits a relatively low IN activity, despite the quite high average platelet thickness of 5.8 nm. Stucki and Tessier (1991) found that SWa-1 in the oxidized states form an extensive network of small parallel-oriented crystals in aqueous suspensions, which may reduce the edge surface available for ice nucleation.

vii.    The synthetic Barasym owes its high IN activity of $F_{het}$ = 0.99 and $T_{het}$ = 245.4 K in a 1 wt % and $F_{het}$ = 0.95 and $T_{het}$ = 243.2 K in a 0.5 wt % suspension to $NH_4^+$ as the charge-balancing cation. Substitution with $Ca^{2+}$ reduces the IN

activity to $F_{het} = 0.61$ and $T_{het} = 239.5$ K in a 0.5 wt% suspension, which is still high in terms of $F_{het}$ but in the typical range for the onset freezing temperature of smectites. With 4.4 unit layers at 100 % RH, the average layer thickness is also in the typical range of smectites investigated by Stepkowska et al. (2004).

viii.   STx-1b exhibited $F_{het} = 0.61 \pm 0.06$ and $T_{het} = 240.3 \pm 0.5$ K for 1 wt % suspensions. The average particle thickness is 4.7 unit layers at 100 % RH. The small lateral dimensions of the platelets (see Table 3) can be a reason for the relatively high heterogeneously frozen fraction.

STx-1b is also the montmorillonite sample that we used for ion exchange experiments, which showed variations in IN activity that cannot be explained by the exchangeable cation. Instead, these differences may be due to differences in the stacking of smectite layers, which may be traced back to the freeze-drying process. Pacula et al. (2006) found that after freeze-drying, the tactoids were more likely to assume a preferred orientation than before this treatment, which might have increased coherent edge surface areas suited to host ice embryos. When the samples were kept over extended periods in water, the decrease in IN activity can be explained by a slowly proceeding delamination in case of $Ca^{2+}$ and $Y^{3+}$ as the exchangeable cation, but not for $Cu^{2+}$. Interestingly, after two years aged in water, the original STx-1b sample seems to consist of a large fraction of fully delaminated particles as $F_{het}$ has reduced to 0.33, but also shows a secondary peak with onset at about $T_{het} \approx 246$ K. We speculate that new sites might have formed due to some aggregation of the delaminated layers over such long time period.

ix.   Finally, freezing on the particles' edges may also explain the low IN activity of partially (K10) and almost completely (KSF) delaminated montmorillonites as observed by Pinti et al. (2012).

## 5. Conclusions and atmospheric implications

The investigation of a series of smectite samples revealed strong differences in IN activity that cannot be explained by the different exchangeable cations and seem unrelated to mineralogical impurities. Yet, it can be related to the stacking and thickness of tactoids. As the basal surfaces should be insensitive to the tactoid stacking, we identify the edges as the location for ice nucleation. Based on the prediction of classical nucleation theory, at least three smectite layers need to be stacked together to reach a sufficiently large area to host a critical ice embryo, and, we hypothesize that the larger the area is, the higher the freezing temperature should be. Comparison with reported platelet thicknesses of the investigated smectite particles suggests that the observed freezing onset temperatures are limited by the surface area provided by the mostly very thin platelets, which may thin even further when suspended in pure water. As this delamination can occur over months, a slowly proceeding delamination of the samples suspended in pure water can explain the decreasing IN activity with time.

Ice nucleation occurring at the edges of clay minerals is in accordance with findings of Klumpp et al. (2023) who explained variations in IN activity of kaolinite and halloysite samples by differences in the edge structures between the samples. All hitherto investigated clay minerals show IN activity in a similar temperature range in emulsion freezing experiments that is enhanced when the samples are suspended in dilute ammonia or ammonium solutions. This points to common structural and

chemical features of their nucleation sites. As the edges are the only invariable surface that clay minerals have in common, they constitute the only locations that can explain these similarities. Furthermore, as the edges are densely hydroxylated, they fulfill a prerequisite for ice nucleation. Hydroxylated aluminosilicate surfaces are a feature that clay minerals have in common with feldspars, whose IN activity also increases in the presence of ammonia and ammonium solutions. Thus, a sufficiently

large area equipped with hydroxy groups that are available for hydrogen bonding to water molecules seems key for ice nucleation by aluminosilicates in general. Besides the precise location of hydroxyl functionalities within the IN active area, additional topographical properties of the edge surface might also influence the quality of the nucleation sites.

The portion of smectites in atmospheric mineral dust composition varies drastically based on origin, source and atmospheric transport (Murray et al., 2012; Boose et al., 2016, Kaufmann et al., 2016). Only a few previous studies have investigated the

IN abilities of smectites, yet, with different smectite samples investigated in different freezing modes, and with different sample aliquots (Hoose et al., 2008; Zimmermann et al., 2008; Welti et al., 2009; Pinti et al., 2012; Atkinson et al., 2013), which hampers the comparison of IN activity between the different studies. Nevertheless, freezing temperatures seem higher in studies where the experiments started with from dry particles. When smectites immersed in water undergo delamination, a decreased IN activity in the immersion mode is indeed expected compared with contact and condensation freezing, which

occur while the samples are wetted. Therefore, the effective IN activity of smectites might be underestimated if the immersion freezing results are also applied to condensation and contact freezing conditions.

Since ice nucleation active sites are typically very few surface features or defects, too small to be easily characterized experimentally, observation of the ice nucleation process on ice active sites is not feasible. In addition, ion exchange and interlayer swelling features of several smectites add another challenge in assessing the underlying ice nucleation mechanism

and the identity of the ice active sites on such surfaces. However, their structural similarities to non-swelling clay minerals provide some indication to similarities in ice nucleation mechanisms. From this study, we suggest that IN ability of smectite particles is limited by platelet thickness. This is in agreement with Klumpp et al. (2023) who elucidate the role of the edges in the IN ability of kaolin minerals. Such aspects of ice nucleation at special sites, e.g. defects and edges, could potentially be explored via molecular simulations which offer the possibility of probing the small spatial dimensions and short timescales

involved in ice nucleation. However, there either lacks detailed atomic configurations for such features for some minerals (e.g. feldspars), or the features in question are currently difficult to generate in simulations to obtain meaningful conclusions (e.g. clays).

**Appendix A: A typical DSC thermogram**

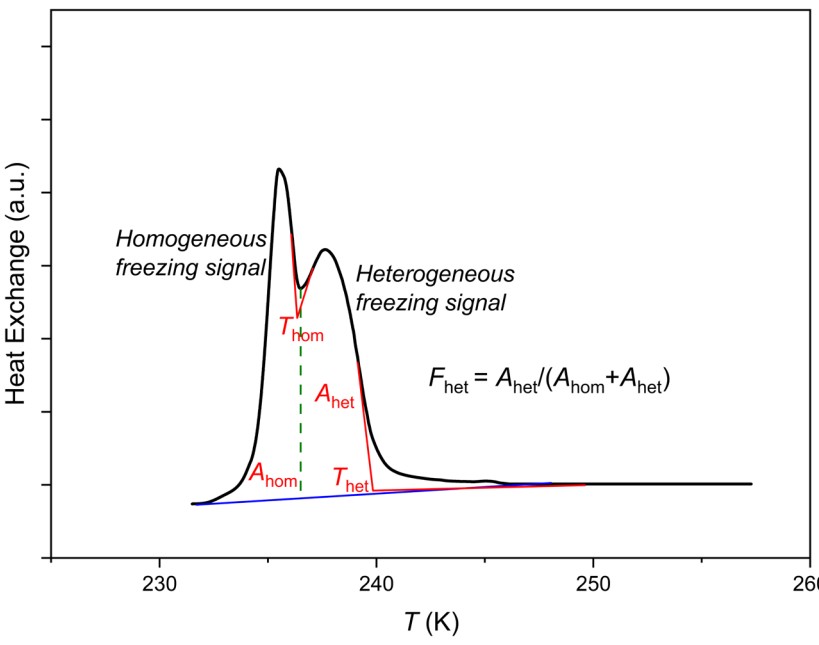


**Figure A.** A typical DSC thermogram showing freezing onset temperatures $T_{het}$ and $T_{hom}$ and the asymptotes used for their construction (red lines), as well as evaluation of $F_{het}$. The area under each peak corresponds to the volume of water that froze homogeneously ($A_{hom}$) or heterogeneously ($A_{het}$). The straight blue line connects the onset of heterogeneous freezing signal with the end of the homogeneous freezing signal and is taken as the base line for evaluating the total frozen fraction. The dashed vertical green line marks the minimum intensity

between the homogeneous and heterogeneous freezing peak and is taken as the separator between the areas under the heterogeneous and homogeneous freezing peaks.

**Appendix B: Semi-quantitative analysis of $NH_4^+$ exchange with $Ca^{2+}$ in Barasym**

A semi-quantitative evaluation of $NH_4^+$ exchange with $Ca^{2+}$ was performed using Nicolet iS10 Fourier Transform Infrared Spectroscopy (FTIR, Nicolet Analytical Instruments, Madison, WI), using a SMART ATR device with a diamond crystal plate

(Thermo Fisher Scientific, Madison, WI) within a range of wavenumbers of 4000 to 500 cm$^{-1}$. This procedure was conducted on original Barasym sample and the $Ca^{2+}$-exchanged Barasym sample. Spectra were recorded at 4 cm$^{-1}$ nominal resolution with mathematical corrections yielding a 1 cm$^{-1}$ actual resolution and averaged value from 50 measurements. Quantification was based on the ratio between the ammonium peak at ~1430 cm$^{-1}$ to that of the structural OH band at ~3620 cm$^{-1}$ as described in (Rytwo et al. 2015). Peak assignments are based on Madejova & Komadel, 2001. Results show that about 30 % of the $NH_4^+$

was replaced.

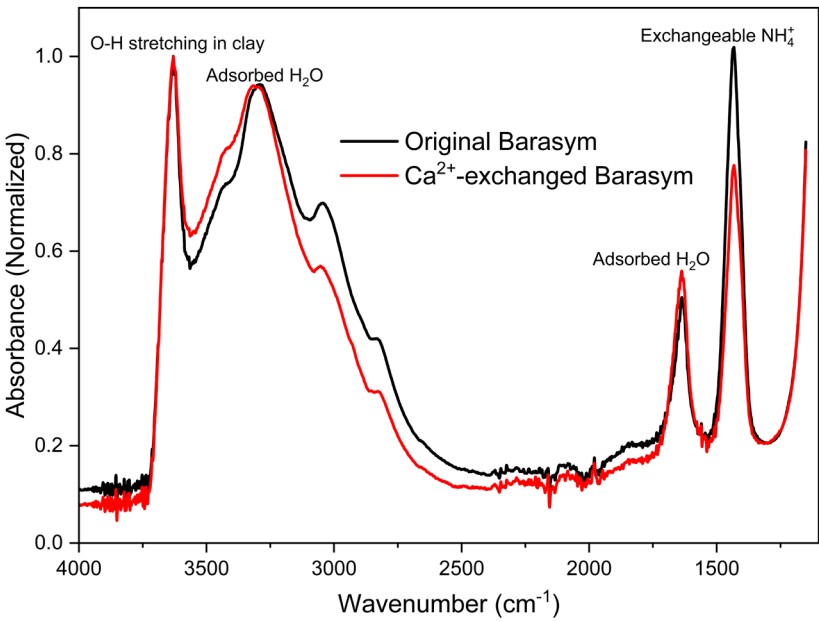

**Figure B.** FTIR spectra for original and $Ca^{2+}$-exchanged Barasym samples for semi-quantitative analysis of $Ca^{2+}$ exchange with $NH_4^+$.


*Data availability.* The data presented in this publication are available at the following repository: https://doi.org/10.3929/ethz-b-000554119.

*Author contributions.* AK, CB and GR conducted the ion exchange experiments. AK and KK conducted the freezing experiments. AK, KK, GR, MP and CM contributed to the planning and interpretation of the experiments. AK and CM prepared the paper with contributions from all co-authors.

*Competing interests.* The contact authors have declared that neither they nor their co-authors have any competing interests.


*Acknowledgements.* We thank Annette Röthlisberger and Marion Rothaupt for the support with the XRD and $N_2$-BET measurements (ClayLab at ETH). We thank Aya Sitruk, Peter Rendel and Shen Levy (all MIGAL Galilee Research Center) for their support during the ion exchange experiments.

*Financial support.* KK has been supported by the Swiss National Foundation (project number 200021_175716).

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
