# Peer review of "Ice nucleation by smectites: The role of the edges"

_EGUsphere, 2022_

## Referee Comment (RC1)

This comment is not the usual full review of the paper under discussion. The minearology aspects and the conclusions of the paper are not questioned. Rather, this comment has a narrow focus on the question of how the DSC measurements described can be interpreted from the point of view of the singular approximation[1]. Two points are addressed: (1) the use of the $T_{het}$ parameter, and (2) the broader question of "surfaces versus sites" as the foci of attention in unraveling the factors governing heterogeneous ice nucleation[2]. The first point is discussed, mainly, as a matter of clarification, but also as a manifestation of the focus of the paper on surfaces, thus emphasizing the importance of the discussion of the second point.

In the paper under discussion, results focus on the observed values of the temperature $T_{het}$, defined by extrapolation of the greatest negative slope of the thermogram to the abscissa. This characterization of the thermogram has been employed in several of the papers cited. It has the benefit of simplicity and allows comparisons of different results to be readily made.

In this paper, and several others using the DSC technique, $T_{het}$ is referred to as the 'onset temperature' of nucleation for the sample being tested. The association of $T_{het}$ with the 'onset' of activity is justified by the fact that $T_{het}$ falls close to the where the thermogram begins to rise above zero. The strictly defined onset, i.e. the first detectable signal, is dependent on instrument sensitivity, emulsion drop sizes and sample concentration. The use of $T_{het}$ as indicator of 'onset' sidesteps those variables and makes it a more robust parameter. However, $T_{het}$ is dependent on the shape of the thermogram which, in addition to the nucleating ability of the sample, is influenced by the drop-size distribution in the emulsion, the size distribution of the suspended particles and perhaps other factors. For identical curves, one could write the simple relationship $T_{het} = T_{ms} + 1.5$, with $T_{ms}$ as the temperature where the slope is steepest and the value of 1.5 is a rough reading from the curves in Fig. 2. In fact, in Fig. 1 the thermograms have quite varied shapes. In Fig. 2, the curves are remarkably similar, but even here, the 5x10-4 curve in panel (a) and the 5x10-2 and 5x10-6 curves in panel (b) do differ to an extent that can be noted by visual comparison of traces of the curves. The influences of these variation on $T_{het}$ are small in Fig. 2 but seem more important for Fig. 5 and can't be judged for the other analyses.

In drop-freezing experiments (with individually observed drops) the onset temperature would be the first freezing event. Most researchers would put little weight on one event because of the uncertainties associated with it. Using the extrapolation notion, the cumulative spectra could be extrapolated to some value near the detection limit of an experiment, or to some selected reference value of $K(T)$ = x. A problem that would arise would be what portion of the $K(T)$ curve to use for the extrapolation, just as the shape of the DSC curve influences the $T_{het}$ point. To complete to comparison, the steepest part of the DSC curve would coincide

1  The singular approximation focuses on the characteristic temperatures of nucleating sites and enumerates the frequency of different sites in terms of differential and cumulative spectra of characteristic temperatures.
2  This 'surfaces versus sites" phrasing of the issue harks back to Vali (2014, Atmos. Chem. Phys. 14, 5271-5294; doi: 10.5194/acp-14-5271-2014).

with a peak in the differential spectrum, provided that variations of droplet sizes in the emulsion is neglected.

The practical issues related to the definition of an onset temperature were elaborated in the preceding paragraphs only to underscore that such a quantity is not a rigorous parameter and would have to be used with some caution.

More important is to consider what an onset temperature, or the steep part of the thermogram, or a peak in the differential spectrum reveal about the sample. Here, one has to consider the totality of the information contained in the thermogram, or in the spectra, i.e. to recognize that nucleation is taking place over a range of temperatures due to nucleating sites of different effectiveness (characteristic temperatures). In this paper the temperature range over which freezing event occur in any given sample is between 3 K and 5 K. That is a relatively narrow range, but comparable to the magnitudes of the differences observed due to surface treatments studied. The range of temperatures is very much dependent on droplet number and particle concentration. In most cases, by varying the concentration and/or by extending the sample size, the range of freezing temperatures can be broadened significantly. This variety of nucleating ability for particles of the same material or same source is the principal challenge and the fundamental interest in the study of heterogeneous ice nucleation.

The authors mention, and seem to agree with the importance of nucleating sites (line 53), yet no further emphsis on site variability is seen in the paper. They do not, as is done in other papers, link the assumption of a uniform surface to stochastic nucleation. They appear to consider that the shifts in the thermograms, as reflected in the shifts in $T_{het}$, are representative of the effects they wish to demonstrate and it is unnecessary to think about what differences among sites may occur on any given type of surface. That argument is perhaps overly simple.

There is a basic disagreement between the approach that searches for physical and chemical characteristics of a surface and its nucleating ability, and the other approach that diagnoses nucleating sites even if the specific features of the site aren't determined. This is not the place to revisit the contrast and the potential overlaps of the two approaches. Briefly, the main advantage of the focus on the surface properties is that those properties can be determined with a large number of observational and theoretical tools. The main handicap for the focus on sites is that there is no direct way to examine the sites; indirect indications are deduced from the observed spectra and the influences of different facotrs; possible configurations of sites are inferred from molecular simulations.

Surface properties undoubtedly set the conditions for site formation; this may lead to some definition of the probability of site formation. However, the variety of sites that form (the range of freezing temperatures) irremediably raise the question of what factors come into play. One has to remember that the sites represent a minuscule fraction of the total surface area and that they differ from one another[3]. Thus, it seems necessary to think about how the
* * *
3    For simplicity of discussion, the singular interpretation is applied in its simplest form. The randomness of embryo formation which leads to time-dependence and to scatter about the characteristic freezing temperature in the actual freezing event are ignored.

types of changes induced by treatments of the surface, as is done in this paper, lead to changes in site occurrence and character. There are attempts in that direction in the paper, but only average surface properties not local features are considered.

The foregoing discussion doesn't contradict the authors' conclusion regarding edge surfaces of smectite plates as most likely nucleation sites. But, for example, ruling out basal surfaces (Section 4.2) mostly on the basis of molecular simulations of surface/water interactions may be softened if local features serving as sites are also considered. It is clear that the results described in the paper don't provide means of identifying what constitute nucleating sites; however, considerations of that problem may lead to the results making a more realistic addition to the accumulating knowledge about heterogeneous ice nucleation.

Minor points, with reference to line numbers:

313-318    The inclusion and exclusion of different samples as active and/or with peaks at 240 K is confusing and not in agreement with what is seen in Fig. 1. Also, what does 'around 240 K' mean?

321-325    The temperatures of 240 K is called a standard freezing peak and those at 246.2 and 247.3 K as special peaks. These temperatures are onset values, according to Fig. 1, not peaks. This is confusing.

323          $T_{het}$ = 246.2 isn't the value shown in Table 4.

~ 345, footnote in Table 4: what is meant by "highest uncertainty of SD" ?

---

## Referee Comment (RC2)

**Review of "Ice Nucleation by smectites: The role of the edges," by A. Kumar et al. 2022**

The preprint "Ice Nucleation by smectites: The role of the edges," is a well written account of freezing experiments of various smectites. In addition to investigating various mineralogies the authors have conducted experiments to examine the role of specific charge carriers (cations) and also how aging in air and water environments might change the ice nucleating potential of these minerals. The work is motivated by observations that various mineral dust particulates are noted to be potentially important atmospheric ice nucleators in certain temperature ranges. While overall I think the manuscript is well written and certainly a quite deep dive into smectite assisted ice formation I do have a few comments and concerns related to the manuscript.

Moreover, as I have not previously evaluated an EGUsphere preprint in this regard I have spent some time trying to ascertain the appropriate way to contextualize the work. My understanding is that this preprint is aimed at ACP audiences. In this case I do think the authors would benefit by bringing their story full circle and reconnecting the work to the atmosphere in the discussion/conclusions. Absent this connection, much of the manuscript reads equally as well as an examination of mineral freezing, absent a strong atmospheric connection. Some questions that the authors might consider revisiting:

I. They mention previous work on illite, and this clay has been (in the context of the INUIT project) proposed as a potential freezing standard that could be used, for example, for instrument inter-comparisons. Do the results presented here shed light on whether such a choice would be useful? For example, any standard might need to be stored for long periods of time with the hope that users one year would observe the same sample characteristics in another.

II. How representative of atmospheric aerosol can one take these emulsion freezing experiments to be? One gets the feeling the 3 part Kumar series cited would need to be re-read in full to appreciate the details. What is the droplet size distribution? What are the biggest and smallest droplets? Why do not all the droplets freeze before homogeneous temperatures? In analogous droplet freezing or other assay experiments, even with quite rarified INP freezing most always proceeds to completion before homogeneous temperatures, and or at higher temperatures than pure water, "blank" experiments.

Perhaps the authors could include a pure water, "blank" DSC thermogram for comparison?

The thermogram measurements themselves are quite interesting but again here I am left wanting a bit more detail. There is reference to the "typical" case and more details to be found in the Kumar et al. 2018 paper. However, in examining Figure 1 of that publication I am left with several questions as it relates to the current study. The methods for peak determination, onset determination, and peak integration laid out in that earlier publication seem to rely a bit on well separated and distinct peaks. Quite often in the study submitted here that is not the case and peaks are more strongly convoluted or sometimes two peaks are not clearly evident. This raises the following questions:

I. Figure 1 of the earlier paper indicates that peak onset is determined using some type of asymptote? Has the same method be used here? Would not a 2nd derivative better reflect the inflection of slope that would indicate onset?

II. When peaks are poorly separated, take many of the traces in Figure 2(b), how are the peaks deconvoluted for integration etc.? It seems that in this case a doublet fitting algorithm would be more suited to the data? What is outlined in the earlier paper seems a bit crude for the convoluted peaks observed here.

III. I understand that the peak normalization $F_{het}+F_{hom}$ implies that all droplets are frozen in all cases (or at least the same amount (volume) of ice is formed in every experiment). Is this also robustly observed? There is no small droplet curvature dependence, that surpresses freezing?

IV. In Figure 1 a few of the traces (SWa-1, 5 wt%, SHCa-1, 5 wt%, Laponite) exhibit one peak only – is this then only homogeneous freezing, as is supposed for the Laponite? Others, e.g. SAu-1, 1 wt%, seem to show three peaks. What is happening in these cases?

V. What physically do $F_{het}$ and $F_{hom}$ represent? I understand that these values are indicative of how likely it is that something freezes heterogeneously vs. homogeneously. However, given these represent normalized integrals of the heat, which I anticipate scales like volume/mass, does this

mean a doubling of F is a $2^3$ increase in heterogeneous freezing? It would be helpful for the authors to give this a meaning that is more easily physically interpreted.

VI. The 2 year trace in Figure 5 contains 2 filled red squares – what does this indicate? Two separate heterogeneous activations?

Point V also ties into the atmospheric applications of the DSC measurements. Is freezing fraction here somehow related to what one might expect for an activated fraction in the atmosphere? That is, beyond onset temperature, how can one translate some of these results at least qualitatively to discuss comparisons with ice nucleation in the atmospheric context. For example, it is understood that soot can nucleate ice, but that it often does so quite inefficiently, like 1 particle per million is an active INP.

Beyond these general questions, below I list some specific questions/comments that are perhaps better considered as the enter into the text.

**Itemized Scientific and Editorial Comments:**

*Specific Suggestions by Page and Line Number (page, line):*

• (1,24) replace "the one" with 'that'

• (Introduction) I would simply like to complement the authors on the very complete mineralogy both here and in the Methodology section.

• (8,232) rephrase, 'is reported as a voltage'

• (Figure 1) It seems this figure could be better utilized with at least freezing onset also indicated.

Also, extracting one of these curves and illustrating the peak deconvolution would be useful.

• (Figure 2) The meaning of "...heterogeneous and homogeneous freezing curves sum up to the same value...." is very vague. The same expression is also used in other captions and also pertains to clearly defining the peak normalization as I have alluded to in point III above. I also recommend using the same terms in the figure and caption. I understand "no solute" and "pure water" to mean the same thing, but it would be better if the phrasing matched.

• (15,366) Although not strictly incorrect, *verbing* a noun like "evidences" is quite often an awkward wording. This verbing of evidence is done multiple times and I would suggest rephrasing.

• (15,391) rephrase, '...no clear trend towards higher or towards lower.....'

A general comment on the "Ion Exchange" section. Have the authors considered how the evolution of the Debye layer and ordering of screening charges may also change water structuring at the interface? I feel that the cation and valency are address, but the near field interactions are not mentioned.

• (16,402) rephrase, '...as made evident...' (see verbing comment above)

• (16,416) rephrase, '...we investigate the relationship between IN activity and particle morphology.'

• (Figure 3,422) 'sum to'

• (Figure 3) Define CEC in main text and utilize here. Not generally appropriate to first define an acronym in a figure caption. SD appears to be used for one standard deviation, but also is not strictly defined. Using '$1\sigma$' would perhaps better communicate the intent if I understand correctly.

• (Figure 3,424) '...a few days...'

***Summation:*** Overall the submitted preprint is quite well written and presents a thorough suite of results related to smectite freezing. To make the manuscript suitable for publication in ACP I believe the above concerns should be addressed. Moreover, the authors should return to the atmosphere in the discussion/conclusions to emphasize the points of connection. Finally, one of the most interesting results is related to edge nucleation being the limiting length scale in the nucleation activity of these materials. However, the evidence for this presented here, while well reasoned, is not direct. Some indication of what kind of direct studies could follow or are planned/underway would add to conclusions.

I think one will find in the freezing literature that not only edges, but steps and basal plane imperfections, like screw dislocations, can initiate freezing. The dimensionality of these may be quite different compared to the layer thickness, and therefore add anchor points that a simple 'edge' model would not have.

–

---

## Author Comment (AC1)

We thank Gabor Vali for his constructive comments. We reproduce his comments in *blue* and our responses in black.

*This comment is not the usual full review of the paper under discussion. The minearology aspects and the conclusions of the paper are not questioned. Rather, this comment has a narrow focus on the question of how the DSC measurements described can be interpreted from the point of view of the singular approximation[1]. Two points are addressed: (1) the use of the $T_{het}$ parameter, and (2) the broader question of "surfaces versus sites" as the foci of attention in unraveling the factors governing heterogeneous ice nucleation[2]. The first point is discussed, mainly, as a matter of clarification, but also as a manifestation of the focus of the paper on surfaces, thus emphasizing the importance of the discussion of the second point.*

*In the paper under discussion, results focus on the observed values of the temperature $T_{het}$, defined by extrapolation of the greatest negative slope of the thermogram to the abscissa. This characterization of the thermogram has been employed in several of the papers cited. It has the benefit of simplicity and allows comparisons of different results to be readily made.*

*In this paper, and several others using the DSC technique, $T_{het}$ is referred to as the 'onset temperature' of nucleation for the sample being tested. The association of $T_{het}$ with the 'onset' of activity is justified by the fact that $T_{het}$ falls close to the where the thermogram begins to rise above zero. The strictly defined onset, i.e. the first detectable signal, is dependent on instrument sensitivity, emulsion drop sizes and sample concentration. The use of $T_{het}$ as indicator of 'onset' sidesteps those variables and makes it a more robust parameter. However, $T_{het}$ is dependent on the shape of the thermogram which, in addition to the nucleating ability of the sample, is influenced by the drop-size distribution in the emulsion, the size distribution of the suspended particles and perhaps other factors. For identical curves, one could write the simple relationship $T_{het} = T_{ms} + 1.5$, with $T_{ms}$ as the temperature where the slope is steepest and the value of 1.5 is a rough reading from the curves in Fig. 2. In fact, in Fig. 1 the thermograms have quite varied shapes. In Fig. 2, the curves are remarkably similar, but even here, the $5x10^{-4}$ curve in panel (a) and the $5x10^{-2}$ and $5x10^{-6}$ curves in panel (b) do differ to an extent that can be noted by visual comparison of traces of the curves. The influences of these variation on $T_{het}$ are small in Fig. 2 but seem more important for Fig. 5 and can't be judged for the other analyses.*

*In drop-freezing experiments (with individually observed drops) the onset temperature would be the first freezing event. Most researchers would put little weight on one event because of the uncertainties associated with it. Using the extrapolation notion, the cumulative spectra could be extrapolated to some value near the detection limit of an experiment, or to some selected reference value of $K(T) = x$. A problem that would arise would be what portion of the $K(T)$ curve to use for the extrapolation, just as the shape of the DSC curve influences the $T_{het}$ point. To complete to comparison, the steepest part of the DSC curve would coincide with a peak in the differential spectrum, provided that variations of droplet sizes in the emulsion is neglected.*

*The practical issues related to the definition of an onset temperature were elaborated in the preceding paragraphs only to underscore that such a quantity is not a rigorous parameter and would have to be used with some caution.*

We agree that it is important to keep in mind the limitations of $T_{het}$ as a measure for ice nucleation (IN) activity. In this work and also in our previous studies, we use $T_{het}$ in combination with $F_{het}$ as a means to quantify and compare IN activities of different samples. Since this is not the full information provided by DSC thermograms, we display and discuss also the whole DSC curves in our studies.

Note, that we mean the whole freezing peak when we refer to a specific $T_{het}$. Therefore, we do not refer to $T_{het}$ as the "onset temperature of nucleation" but as the "freezing onset temperature". With "freezing", we mean the process of ice nucleation and growth, which gives rise to the heat signal in the DSC. Due to the negligible heat release associated with nucleation, nucleation alone is invisible in DSC. The same is true for the freezing of a single micrometer-sized droplet. We use the freezing onset instead of the freezing peak temperature to characterize the position of the freezing peak because the location of the peak maximum is influenced by the total heat release (see Marcolli et al., 2007) and therefore is a less robust parameter than the onset temperature to reference a DSC peak. We have a detailed discussion in Section 2.4.1.

*More important is to consider what an onset temperature, or the steep part of the thermogram, or a peak in the differential spectrum reveal about the sample. Here, one has to consider the totality of the information contained in the thermogram, or in the spectra, i.e. to recognize that nucleation is taking place over a range of temperatures due to nucleating sites of different effectiveness (characteristic temperatures). In this paper the temperature range over which freezing event occur in any given sample is between 3 K and 5 K. That is a relatively narrow range,*

*but comparable to the magnitudes of the differences observed due to surface treatments studied. The range of temperatures is very much dependent on droplet number and particle concentration. In most cases, by varying the concentration and/or by extending the sample size, the range of freezing temperatures can be broadened significantly. This variety of nucleating ability for particles of the same material or same source is the principal challenge and the fundamental interest in the study of heterogeneous ice nucleation.*

*The authors mention, and seem to agree with the importance of nucleating sites (line 53), yet no further emphasis on site variability is seen in the paper. They do not, as is done in other papers, link the assumption of a uniform surface to stochastic nucleation. They appear to consider that the shifts in the thermograms, as reflected in the shifts in $T_{het}$, are representative of the effects they wish to demonstrate and it is unnecessary to think about what differences among sites may occur on any given type of surface. That argument is perhaps overly simple.*

The underlying assumption of our study is that freezing occurs on nucleation sites, which require a minimum size to host a critical ice embryo, and that this size determines the nucleation temperature. On a specific site, nucleation occurs with a certain probability that increases with decreasing temperature and can be described by classical nucleation theory as discussed in Kaufmann et al. (2017). As the width of the freezing signal in DSC is determined by heat release and not by the temperature distribution at which nucleation occurs, the signal cannot be directly related to nucleation. Moreover, as the heat signal per nucleation event depends on the droplet size, an exact knowledge of the droplet size distribution is required to extract nucleation temperatures from the signal. We undertook such a detailed analysis in Marcolli et al. (2007). In the subsequent studies we concentrated more on comparison of immersion freezing in the presence of solutes, for which DSC is well suited as it allows the measurement of different solutes and solute concentrations within an acceptable amount of time.

*There is a basic disagreement between the approach that searches for physical and chemical characteristics of a surface and its nucleating ability, and the other approach that diagnoses nucleating sites even if the specific features of the site aren't determined. This is not the place to revisit the contrast and the potential overlaps of the two approaches. Briefly, the main advantage of the focus on the surface properties is that those properties can be determined with a large number of observational and theoretical tools. The main handicap for the focus on sites is that there is no direct way to examine the sites; indirect indications are deduced from the observed spectra and the influences of different factors; possible configurations of sites are inferred from molecular simulations.*

*Surface properties undoubtedly set the conditions for site formation; this may lead to some definition of the probability of site formation. However, the variety of sites that form (the range of freezing temperatures) irremediably raise the question of what factors come into play. One has to remember that the sites represent a minuscule fraction of the total surface area and that they differ from one another[3]. Thus, it seems necessary to think about how types of changes induced by treatments of the surface, as is done in this paper, lead to changes in site occurrence and character. There are attempts in that direction in the paper, but only average surface properties not local features are considered.*

We fully agree. We are confronted with the fundamental problem that surface specific characterization methods lack the required resolution to resolve nucleation sites. Therefore, specific surface features need to be derived from average sample properties. The original idea behind this study was to investigate how IN activity depends on the exchangeable cations. To our surprise we found no such dependence, but we were astonished by the high variability of the heterogeneous freezing signal between smectite samples, which we tried to explain in the following.

*The foregoing discussion doesn't contradict the authors' conclusion regarding edge surfaces of smectite plates as most likely nucleation sites. But, for example, ruling out basal surfaces (Section 4.2) mostly on the basis of molecular simulations of surface/water interactions may be softened if local features serving as sites are also considered. It is clear that the results described in the paper don't provide means of identifying what constitute nucleating sites; however, considerations of that problem may lead to the results making a more realistic addition to the accumulating knowledge about heterogeneous ice nucleation.*

Our discussion of ice nucleation activity relies on the assumption that ice nucleation occurs on rare sites that provide suitable properties for ice embryo formation. On several occasions in the manuscript, we refer to nucleation sites, and in Sect. 4.5, we specify their approximate size as derived from classical nucleation theory and molecular simulations.

*Minor points, with reference to line numbers:*

*313-318 The inclusion and exclusion of different samples as active and/or with peaks at 240 K is confusing and not in agreement with what is seen in Fig. 1. Also, what does 'around 240 K' mean?*

We agree that the presence of a peak around 240 K as a common feature of the DSC curves does not become obvious at first glance. However, evaluation of the onset freezing temperatures reveals a $T_{het}$ value between 239.9 and 241.2 K for all samples with the exception of Barasym and SHCa-1.We make this clearer in the revised manuscript by referring explicitly to Table 4, where we list $T_{het}$, and by exactly giving the freezing range in Section 3.1: "All IN active smectites with the exception of Barasym and the hectorite SHCa-1 exhibit a freezing peak with $T_{het}$ around 240 K (exact range of 239.9–241.2 K, see Table 4)."

*321-325 The temperatures of 240 K is called a standard freezing peak and those at 246.2 and 247.3 K as special peaks. These temperatures are onset values, according to Fig. 1, not peaks. This is confusing.*

As explained above, we characterize freezing peaks by their onsets rather than the peak maxima because the onset is the more robust parameter than the peak maximum. With $T_{het}$, we refer to the full peak.

*323 $T_{het} = 246.2$ isn't the value shown in Table 4.*

Thank you for pointing this out. We have corrected the values in the revised manuscript.

*~ 345, footnote in Table 4: what is meant by "highest uncertainty of SD" ?*

We meant the largest uncertainty across all investigated samples. We have changed the footnote to "uncertainty does not exceed 0.5 K and 0.1 for $T_{het}$ and $F_{het}$, respectively."
* * *
*1 The singular approximation focuses on the characteristic temperatures of nucleating sites and enumerates the frequency of different sites in terms of differential and cumulative spectra of characteristic temperatures.*

*2 This 'surfaces versus sites" phrasing of the issue harks back to Vali (2014, Atmos. Chem. Phys. 14, 5271- 5294; doi: 10.5194/acp-14-5271-2014).*

*3 For simplicity of discussion, the singular interpretation is applied in its simplest form. The randomness of embryo formation which leads to time-dependence and to scatter about the characteristic freezing temperature in the actual freezing event are ignored.*

---

## Author Comment (AC2)

We thank reviewer 2 for the constructive comments. We reproduce reviewer's comments in *blue* and our responses in black.

*The preprint "Ice Nucleation by smectites: The role of the edges," is a well written account of freezing experiments of various smectites. In addition to investigating various mineralogies the authors have conducted experiments to examine the role of specific charge carriers (cations) and also how aging in air and water environments might change the ice nucleating potential of these minerals. The work is motivated by observations that various mineral dust particulates are noted to be potentially important atmospheric ice nucleators in certain temperature ranges. While overall, I think the manuscript is well written and certainly a quite deep dive into smectite assisted ice formation I do have a few comments and concerns related to the manuscript.*

*Moreover, as I have not previously evaluated an EGUsphere preprint in this regard I have spent some time trying to ascertain the appropriate way to contextualize the work. My understanding is that this preprint is aimed at ACP audiences. In this case I do think the authors would benefit by bringing their story full circle and reconnecting the work to the atmosphere in the discussion/conclusions. Absent this connection, much of the manuscript reads equally as well as an examination of mineral freezing, absent a strong atmospheric connection. Some questions that the authors might consider revisiting:*

*I. They mention previous work on illite, and this clay has been (in the context of the INUIT project) proposed as a potential freezing standard that could be used, for example, for instrument intercomparisons. Do the results presented here shed light on whether such a choice would be useful? For example, any standard might need to be stored for long periods of time with the hope that users one year would observe the same sample characteristics in another.*

Illites have cation exchange capacity higher than that of kaolinite but lower than that of swelling smectites (Kahr and Madsen, 1995). Several layers of water adsorption have been reported for illites when exposed to high RH conditions (Schuttlefield et al., 2007; Baltrusaitis and Grassian, 2009). This can be put into context with storing samples over long time periods. Given the robustness of illite (as well as kaolinite) in terms of negligible ion exchange and interlayer swelling, it seems to be a good freezing standard. For example, the same illite NX sample tested via emulsion freezing experiments by Pinti et al. (2012) and Kaufmann et al. (2016), albeit with a time gap of few years, does not show any remarkable difference in heterogeneous freezing onset temperature.

*II. How representative of atmospheric aerosol can one take these emulsion freezing experiments to be? One gets the feeling the 3-part Kumar series cited would need to be re-read in full to appreciate the details. What is the droplet size distribution? What are the biggest and smallest droplets? Why do not all the droplets freeze before homogeneous temperatures? In analogous droplet freezing or other assay experiments, even with quite rarified INP freezing most always proceeds to completion before homogeneous temperatures, and or at higher temperatures than pure water, "blank" experiments. Perhaps the authors could include a pure water, "blank" DSC thermogram for comparison?*

For clarity, we have this discussion in the Methodology (Section 2.4.1). As suggested by the reviewer, we have updated Figure 1 and included a "pure water" DSC thermogram (orange curve) for comparison. "The median droplet diameter in the emulsion is ~2–3 µm in terms of number size distribution, but a broad distribution in terms of volume. Droplets with diameters of about 12 µm are considered to be relevant for the freezing onset in the DSC experiments (Marcolli et al., 2007; Kaufmann et al., 2016). The random spikes at temperatures warmer than the heterogeneous freezing onset are caused by the freezing of few large droplets (sometimes up to 300 µm present at the tail-end of the droplet size distribution) and are excluded from the evaluation. Their presence is likely due to the coalescence of some smaller droplets probably while transferring the sample to the aluminum pan for DSC and is not reproducible.

In case of emulsions with pure water only ("blank water"), the droplets start to freeze at 237±0.2 K (Figure 1). However, when we introduce dust particles, the number of particles in a single droplet is governed by the volume of the droplet. With increasing droplet volume, the probability of accommodating particles in that droplet increases. Hence, the freezing of larger droplets dominates the heterogeneous freezing signal. While, the homogeneous freezing signal results either from the freezing of smaller empty water droplets or droplets containing particles which are ice nucleation inactive. $F_{het}$ reported in this study cannot be directly translated into absolute quantifiable parameters, rather it should be considered as a qualitative parameter to compare the efficacy of ice nucleation of different dust samples or to assess the changes in efficacy due to different treatments. We also use heterogeneous freezing onset to characterize the freezing temperature because it is a very well-defined parameter easily evaluated from the thermograms. A combination of $T_{het}$ and $F_{het}$ parameters provide a good measure of the overall ice nucleation abilities of the smectites."

Several droplet freezing setups or other assay experiments (e.g. Yun et al. (2020), Whale et al. (2018)) utilize droplets that are several orders of magnitude higher in volume and are in contact with a flat hydrophobic surface. Such few and large droplets freeze entirely due to a single ice nucleation event and represent the best, yet rare, ice nucleation active sites on a given surface. In comparison, the emulsion freezing experiments yield better statistics by utilizing large populations of small droplets in a single experiment. However, DSC does not register freezing event but the heat flow associated with the heat release during freezing.

The relationship between freezing events and heat flow has been shown in Marcolli et al. (2007). This imparts another challenge in directly comparing droplet assays and emulsion freezing experiments.

As mentioned previously, an emulsion primarily consists of droplets of diameter less than 12 μm, with varying numbers of dust particles. The assessment of particle surface area exposed in a single droplet is a challenge owing to the size distributions of both the droplets and the dust particles. Therefore, emulsions are not entirely representative of the aqueous cloud droplets (typically 5–50 μm in diameter). However, the supercooled temperature range that can be probed using emulsion freezing is of utmost relevance for mixed-phase cloud regime.

*The thermogram measurements themselves are quite interesting but again here I am left wanting a bit more detail. There is reference to the "typical" case and more details to be found in the Kumar et al. 2018 paper. However, in examining Figure 1 of that publication I am left with several questions as it relates to the current study. The methods for peak determination, onset determination, and peak integration laid out in that earlier publication seem to rely a bit on well separated and distinct peaks. Quite often in the study submitted here that is not the case and peaks are more strongly convoluted or sometimes two peaks are not clearly evident. This raises the following questions:*

*I. Figure 1 of the earlier paper indicates that peak onset is determined using some type of asymptote? Has the same method be used here? Would not a 2nd derivative better reflect the inflection of slope that would indicate onset?*

We define the freezing temperatures ($T_{het}$ and $T_{hom}$) as the onset points of the freezing signals (i.e., intersection of the tangents at the greatest slope of the freezing signal (intersection of the tangents at the greatest slope of the freezing signal and the extrapolated baseline) as outlined in Figure 1 of Kumar et al. (2018) (also, Figure 2 of Klumpp et al. (2022b)). Indeed, a $2^{nd}$ derivative would yield a proper inflection of slope. However, the uncertainties in $T_{het}$ and $T_{hom}$ observed via multiple freezing runs of any sample encompass that arising from inflection point variations in a narrow temperature range. For clarity, we added a Figure A in the Appendix A, which illustrates the peak onset and frozen volume fraction derivations.

*II. When peaks are poorly separated, take many of the traces in Figure 2(b), how are the peaks deconvoluted for integration etc.? It seems that in this case a doublet fitting algorithm would be more suited to the data? What is outlined in the earlier paper seems a bit crude for the convoluted peaks observed here.*

A vertical line drawn from the minimum intensity point between the homogeneous and heterogeneous freezing peak is taken as the separator between the areas under the heterogeneous and homogeneous freezing peaks. We consider this signal separation method as more robust than curve deconvolution considering the variability in curve shapes, though it is crude and comes with considerable uncertainties (Figure A in revised manuscript). Nevertheless, the combination of $T_{het}$ and $F_{het}$ gives a good overview of the discernable changes in ice nucleation ability due to ion exchange and aging conditions.

*III. I understand that the peak normalization $F_{het}+F_{hom}$ implies that all droplets are frozen in all cases (or at least the same amount (volume) of ice is formed in every experiment). Is this also robustly observed? There is no small droplet curvature dependence, that suppresses freezing?*

The total volume of water in each emulsion sample is not exactly the same. This is due to volume displacement by the added dust particles (which vary in size in each batch) in every aliquot of suspension taken to prepare an emulsion. The heat released during freezing is approximately proportional to the volume of water that freezes in the sample with a minor deviation from this proportionality arising from the temperature dependence of the freezing enthalpy (Marcolli et al., 2007). Therefore, the latent heat released during a freezing experiment varies from emulsion-to-emulsion. Hence, evaluating $F_{het}$ as a normalized parameter (i.e. ratio of the heterogeneous freezing signal to the total freezing signal) helps to compare the general ice nucleation behavior of the samples across multiple emulsion tests.

*IV. In Figure 1 a few of the traces (SWa-1, 5 wt%, SHCa-1, 5 wt%, Laponite) exhibit one peak only – is this then only homogeneous freezing, as is supposed for the Laponite? Others, e.g. SAu-1, 1 wt%, seem to show three peaks. What is happening in these cases?*

Contrary to Laponite, the SWa-1 and SHCa-1 samples show heat signals that are not entirely flat baselines before reaching their homogeneous freezing signals. Hence, we report in Table 4 a heterogeneous freezing signal for SWa-1 and SHCa-1, albeit very faint and flat. The weak signals are difficult to visualize when DSC curves are stacked one on top of the other. For clarity, we have added markers to highlight the heterogeneous freezing onset points in Figure 1.

Indeed, SAu-1 and MX-80 show two distinct heterogeneous freezing signals, with $T_{het}$ values of 247/240 K and 247/241 K, respectively. The dual signals suggest activation of different sets of ice nucleation active sites on such surfaces in two different temperature regimes. Though, we think that both types of sites belong to smectites' surface features, instead of impurities. Our reasoning for this is discussed in detail towards the end of Section 3.1.

*V. What physically do $F_{het}$ and $F_{hom}$ represent? I understand that these values are indicative of how likely it is that something freezes heterogeneously vs. homogeneously. However, given these represent normalized integrals of the heat, which I anticipate scales like volume/mass, does this mean a doubling of F is a $2^3$ increase in heterogeneous freezing? It would be helpful for the authors to give this a meaning that is more easily physically interpreted.*

We have clarified some of these aspects in point III above. To add to that, DSC does not register freezing events but the heat flow associated with the heat release during freezing. As previously mentioned, the relationship between freezing events and heat flow has been shown in Marcolli et al. (2007). The heat release is approximately proportional to the volume of water that froze heterogeneously or homogeneously and is represented by the integral of the peak over time (not temperature). Several droplet assay studies report the ice nucleation active site density ($n_s$), assuming the deterministic description for nucleation events, to compare ice nucleation behavior of variety of particles. This is rather convenient to translate droplet number frozen fractions to $n_s$, due to monodisperse distribution of droplets (Vali et al., 2015; Whale et al., 2018; Yun et al., 2020). On the other hand, it is a drawback of the DSC method that droplet number frozen fractions cannot be derived directly. We attempted to quantify this in Kaufmann et al. (2016), but significant uncertainties were observed which are associated with the dust particle size distribution, the droplet size distribution and frozen water volume fractions in cases with overlapping heterogeneous and homogeneous freezing signals. Therefore, we use water volume frozen fractions ($F_{het}$) as parameter to assess the changes in general efficacy of dust samples to nucleate ice. For example, any changes observed in $F_{het}$ (in combination with $T_{het}$) for a sample are indicative of general effect of a treatment (ion exchange and/or adsorption, aging, surface dissolution, etc.) on its overall ice nucleation ability.

*VI. The 2-year trace in Figure 5 contains 2 filled red squares – what does this indicate? Two separate heterogeneous activations?*

Indeed. We ran freezing experiments on two different STx-1b suspensions that had been aged for two years, and both showed a secondary freezing signal at 246 K. The origin of this signal is not clear. We speculate that new sites might have formed due to some aggregation of the delaminated layers over such long time period. We have addressed this in Section 4.5(viii).

*Point V also ties into the atmospheric applications of the DSC measurements. Is freezing fraction here somehow related to what one might expect for an activated fraction in the atmosphere? That is, beyond onset temperature, how can one translate some of these results at least qualitatively to discuss comparisons with ice nucleation in the atmospheric context. For example, it is understood that soot can nucleate ice, but that it often does so quite inefficiently, like 1 particle per million is an active INP.*

As we mentioned previously, $F_{het}$ reported in this study cannot be directly translated into absolute quantifiable parameters, rather it should be considered as a qualitative parameter to compare the efficacy of ice nucleation of different dust samples or to assess the changes in efficacy due to different treatments. We also use heterogeneous freezing onset to characterize the freezing temperature because it is a very well-defined parameter easily evaluated from the thermograms. The range of freezing temperature observed for various smectites are in agreement with previous studies carried out with similar dust loadings of water droplets (Welti et al., 2009; Atkinson et al., 2013). A combination of $T_{het}$ and $F_{het}$ parameters provide a good measure of the overall ice nucleation abilities of the smectites. As suggested by the reviewer, we add a discussion on atmospheric implications to Section 5:

"The portion of smectites in atmospheric mineral dust composition varies drastically based on origin, source and atmospheric transport (Murray et al., 2012; Boose et al., 2016; Kaufmann et al., 2016). Only few previous studies have investigated the IN abilities of smectites, yet, with different smectite samples investigated in different freezing modes, and with different sample aliquots (Hoose et al., 2008; Zimmermann et al., 2008; Welti et al., 2009; Pinti et al., 2012; Atkinson et al., 2013), which hampers the comparison of IN activity between the different studies. Nevertheless, freezing temperatures seem higher in studies where the experiments started from dry particles. When smectites immersed in water undergo delamination, a decreased IN activity in immersion mode is indeed expected compared with contact and condensation freezing, which occur while the samples are wetted. Therefore, the effective IN activity of smectites might be underestimated if immersion freezing results are also applied to condensation and contact freezing conditions.

Since ice nucleation active sites are typically very few surface features or defects, too small to be easily characterized experimentally, observation of the ice nucleation process on ice active sites is not feasible. In addition, ion exchange and interlayer swelling features of several smectites add another challenge in assessing the underlying ice nucleation mechanism and the identity of the ice active sites on such surfaces. However, their structural similarities to non-swelling clay minerals provide some indication to similarities in ice nucleation mechanisms. From this study, we suggest that IN ability of smectite particles is limited by platelet thickness. This is in agreement with Klumpp et al. (2022a) who elucidate the role of the edges in the IN ability of kaolin minerals."

*Beyond these general questions, below I list some specific questions/comments that are perhaps better considered as the enter into the text.*

*Itemized Scientific and Editorial Comments:*

*Specific Suggestions by Page and Line Number (page, line):*

- *(1,24) replace "the one" with 'that'*
  Done.
- *(Introduction) I would simply like to complement the authors on the very complete mineralogy both here and in the Methodology section.*
  We thank the reviewer for the appreciation.
- *(8,232) rephrase, 'is reported as a voltage'*
  Done.
- *(Figure 1) It seems this figure could be better utilized with at least freezing onset also indicated. Also, extracting one of these curves and illustrating the peak deconvolution would be useful.*
  We have revised the figure as suggested by the reviewer. Additional figure showing $T_{het}$ and $F_{het}$ evaluation is added in the Appendix A.
- *(Figure 2) The meaning of "...heterogeneous and homogeneous freezing curves sum up to the same value...." is very vague. The same expression is also used in other captions and also pertains to clearly defining the peak normalization as I have alluded to in point III above. I also recommend using the same terms in the figure and caption. I understand "no solute" and "pure water" to mean the same thing, but it would be better if the phrasing matched.*
  We have changed the expression to "All curves are normalized with respect to the total areas under the heterogeneous and homogeneous freezing curves." Terminology corrected in the captions for pure water cases (replaced with the name of the sample), as suggested by the reviewer.
- *(15,366) Although not strictly incorrect, verbing a noun like "evidences" is quite often an awkward wording. This verbing of evidence is done multiple times and I would suggest rephrasing.*
  Done.
- *(15,391) rephrase, '...no clear trend towards higher or towards lower.....'*
  Done.

*A general comment on the "Ion Exchange" section. Have the authors considered how the evolution of the Debye layer and ordering of screening charges may also change water structuring at the interface? I feel that the cation and valency are address, but the near field interactions are not mentioned.*

We have renamed Section 4.1 to "Exchangeable cations and ion adsorption" and added the following discussion.

"It has been previously established that feldspars can undergo cation exchange and adsorption when exposed to salt solutions (Nash and Marshall, 1957; Demir et al., 2001; Demir et al., 2003). Contrary to laboratory experiments, classical molecular dynamics (MD) simulations have not been able to capture ice nucleation on K-feldspar surfaces. Though (001), (010) and (100) surfaces of K-feldspar exposed to various salt solutions (including ammonium-containing solutions) show that sorbed ions affect interfacial water orientation, albeit unfavorable for ice nucleation (Kumar et al., 2021). MD simulations performed on basal surface of kaolinite exposed to various solutions revealed that the interfacial ions played minor part in inhibiting ice nucleation by blocking the regions of clear surface (Ren et al., 2020). On the other hand, our study shows that within measurement uncertainty, the IN ability of fresh STx-1b is not affected when cation concentrations are close to or even surpass the CEC limit where surface and interfacial ion concentrations might be high."

- *(16,402) rephrase, '...as made evident...' (see verbing comment above)*
  Done.
- *(16,416) rephrase, '...we investigate the relationship between IN activity and particle morphology.'*
  Done.
- *(Figure 3,422) 'sum to'*
  Done.
- *(Figure 3) Define CEC in main text and utilize here. Not generally appropriate to first define an acronym in a figure caption. SD appears to be used for one standard deviation, but also is not strictly defined. Using '1' would perhaps better communicate the intent if I understand correctly.*
  Done
- *(Figure 3,424) '...a few days...'*
  Done

*Summation: Overall the submitted preprint is quite well written and presents a thorough suite of results related to smectite freezing. To make the manuscript suitable for publication in ACP I believe the above concerns should be addressed. Moreover,*

*the authors should return to the atmosphere in the discussion/conclusions to emphasize the points of connection. Finally, one of the most interesting results is related to edge nucleation being the limiting length scale in the nucleation activity of these materials. However, the evidence for this presented here, while well-reasoned, is not direct. Some indication of what kind of direct studies could follow or are planned/underway would add to conclusions. I think one will find in the freezing literature that not only edges, but steps and basal plane imperfections, like screw dislocations, can initiate freezing. The dimensionality of these may be quite different compared to the layer thickness, and therefore add anchor points that a simple 'edge' model would not have.*

We have added the following discussion in Section 5 to address these aspects.

[revised manuscript text omitted]

---

## Referee Report (RR1)

**Follow-up review of "Ice Nucleation by smectites: The role of the edges," by A. Kumar et al. 2022**

I feel the authors have adequately addressed my concerns. However, I would still argue that the way the authors treat the DSC curve analysis is rather rough and perhaps more quantitative data could emerge if better curve fitting algorithms were employed. Although we have not made the same suggestions I see from the Vali response that he had similar ideas. Moreover, it seems in the response to his point that a peak in the differential spectrum of the DSC curve would correspond precisely with the location of the steepest slope and that might be another useful marker seemed to be a bit misinterpreted. This is not peak temperature, as the response seems to narrow in on. Finally because the shapes of the curves (i.e., the tails) will not always be the same for the *het* and *hom* cases I do not entirely buy the argument that the chosen method of integration would not be significantly outperformed by a doublet deconvolution, and thus perhaps yield more insight. All of the singly peaked DSC curves are quite symmetrical, why would we not expect the same of others? Thus knowing one tail should allow robust fitting of the other (the part convoluted with the other freezing behavior).

This said, the author's larger point is that this information leads only to relative weighting and this is only one of a few barriers in the technique that prevent quantified nucleation information from emerging. Rather the point is more to give insight into what promotes freezing earlier versus later.

**Itemized Scientific and Editorial Comments from the resubmitted tracked changes manuscript version:**

*Specific Suggestions by Page and Line Number (page, line):*

- (11,293) should read....and are also shown in....

- (11,294) Rigorously $\sim$ means similar to, as in the same order of magnitude, whereas $\approx$ means approximately. Here the later should be used given the range is less than 1 order of magnitude.

- (11,300) should read....In the case of emulsions formed with pure water....

- (Figure 1, caption) should read.....For references purposes a DSC....for the samples.

- (22,487) The 2 sentences beginning, "Though (001)," and ending..."(Ren et al., 2020)" are difficult to read and include dangling clauses etc. I would suggest rephrasing to improve clarity.

- (30,673) should read....Only a few....

- (30,677) should read...started with dry particles

- (30,689) should read......atomic configurations for ....

- (30,690) should read...in simulations to obtain....

A general comments: at times the usage of "in immersion freezing mode" seems like it would better written ....in the immersion freezing mode.

Although willing to review another edition of this manuscript, I do not see any acute need. I can recommend the manuscript for publication when these technical corrections are made and the editor is happy.

---

## Author Response (AR2)

We thank Gabor Vali for his constructive comments. We reproduce his comments in *blue* and our responses in black.

*Although it may be marginal to the specific topic of this paper, it seems worthwhile to pursue further points raised in my previous comment about how DSC results may be viewed with a sharper focus on the nucleation events which are to be studied. DSC data make important contributions to ice nucleation studies and the opportunity arose with this paper to discuss some aspects of those measurements with a view toward better exchange of information between different researchers.*

Gabor Vali raises here important questions concerning the interpretation of freezing experiments performed with DSC. We consider these points as a general discussion of the potential of DSC to investigate ice nucleation and we will therefore formulate our responses accordingly.

*The paper is consistent with the terminology it uses (as defined on lines 287-290 and in Fig. A) in accord with previous publications. To this reviewer, that terminology has some unfortunate aspects. Strictly speaking, there is no 'heterogeneous and homogeneous freezing' and it is not a good shorthand for indicating heterogeneous and homogeneous nucleation events. Similarly, 'freezing peak' is an awkward shorthand designation. It is understandable, when describing DSC data, that freezing is used in many contexts since that is the source of the detected signal. However, since the goal of the measurements is to diagnose and understand nucleation, that usage is distracting. Reference to freezing diverts attention and loses the sharpness that nucleation implies as a near-instantaneous process at a very specific location.*

The distinction between ice nucleation and freezing is crucial in discussing our DSC results. During DSC experiments with emulsion samples, freezing of each droplet in the emulsion is initiated by a nucleation event. In contrast, the heat signal just informs us about the heat release during freezing, which is delayed with respect to ice nucleation and droplet freezing as discussed in Marcolli et al. (2007). Even if nucleation resulted in complete, instantaneous freezing, we would not be able to derive nucleation events from the DSC signal, because the delay in heat transfer is substantial. The analysis in Marcolli et al. (2007) indicates that in our DSC instrument, the heat release of each droplet follows an exponential decay law with a time constant of 11 s. Most importantly, the heat retention transiently increases the DSC sample temperature, resulting in a temporary deviation from the nominal cooling rate. Depending on the intensity of the heat release, nucleation might be almost complete when the heat signal peaks. Therefore, the onset is the best-defined temperature to reference a heat signal.

*The foregoing point comes into focus with respect to Thet. I raised questions about the physical meaning of this parameter in my previous comment. The authors responded by saying that they do not mean it to be a threshold or onset of nucleation, as I referred to it, but a 'freezing onset temperature'. Since there is no freezing without nucleation this response eludes the real issue of the interpretation of what Thet stands for. Onset is not a good way to characterize the nucleating abilities of samples in general, because it arises from a small number of events highly dependent on particle concentration (loading) and chance. With Thet the meaning of onset is even more fuzzy are there are evident DSC signals beyond Thet indicating some nucleation activity of the sample at higher temperatures. So, while Thet is a convenient reference temperature, calling it something else than 'onset' temperature would avoid questions about its meaning. By determining Thet by extrapolation from the slope of the thermal signature, the problem of dependence on a small fraction of the nucleation events is avoided but the ambiguity of the physical meaning is not resolved.*

*Perhaps consideration has been already given to using the point of steepest slope of the thermal signal as the reference temperature. That temperature, say Ts corresponds to large numbers of nucleation events, albeit not necessarily a maximum, and it seems easier to relate to than to Thet. Neglecting droplet size*

*variations, or in cases where uniform droplets are used, Ts would correspond to a peak in the differential nucleus spectrum. The use of Ts would avoid awkward situations such as having Thet appear on a totally flat part of the curve displayed, e.g. SHCa-1. 5% in Fig. 1. which is probably due to the normalization of the curves to a uniform height making the first pulse inappreciable.*

With $T_{\text{het}}$, we refer to the onset of the freezing peak recorded by DSC. The onset temperature of a DSC signal is an unambiguous parameter that we apply based on its general definition within the field of DSC. Using $T_s$ as proposed by Gabor Vali would not change the appearance of the onset being on the flat part of the curve for e.g. SHCA-1. We have added DSC thermograms of SHCa-1, SWa1 and Laponite samples separately in Fig. 1(b) for better clarity.

$T_{\text{het}}$ should not be identified with the onset of ice nucleation. The onset of ice nucleation, although frequently used in ice nucleation studies, is in fact an ill-defined parameter as it refers to specific setups that are characterized by a sample volume in which the INP number or mass are usually only varied within narrow limits. Because continuous flow diffusion chambers investigate a much lower number of INPs, the freezing onset of an INP type in these instruments is much lower than in a drop-freezing assay with microliter-sized sample aliquots. Owing to the small droplet size in emulsions, ice nucleation temperatures in these experiments are in a similar range as the ones in continuous-flow diffusion chambers and much lower than the ones in drop-freezing assays.

In emulsion freezing experiments, the INP surface or mass per droplet is subject to large uncertainties because of the polydispersity of the emulsion droplets as discussed in the appendix of Kaufmann et al. (2016). Because of the uncertainties in nucleation temperatures and amount of INPs per droplet, emulsion freezing experiments performed with DSC are not suited to derive INP spectra or ice-nucleation active site densities.

*The other parameter used to characterize nucleation activity in a sample is Fhet. This parameter is a measure of the mass loading of the sample and the size distribution of particles in the droplets with the nucleating material. Beyond comparisons, it provides little quantitative information on the nucleating ability of the material being tested.*

We are well aware of this drawback. The strength of DSC is the capability to compare the ice-nucleation activity of different samples with reasonable expenditure of time. To build on the strength of our method, we focus in our DSC emulsion freezing experiments on the comparison of similar samples, like the comparison of the IN activity of different smectites in this study, or changes in ice-nucleation activity due to the addition of solutes. Comparison with ice nucleation data obtained with other techniques would require the transformation to INP spectra, which is not feasible with DSC emulsion freezing experiments as explained above. We think that microfluidic devices would be the instruments of choice to provide ice nucleation data that can be converted to INP spectra in the temperature range below 250 K.

*In essence, the foregoing comments are aimed at shifting focus in the descriptions of the results on nucleation rather than the freezing that results from it. No specific recommendations are made here on how to achieve that. The authors and the editor may decide that this is not the time to reconsider how DSC data are presented.*

We fully agree that a shift from freezing to ice nucleation would be desirable. Moreover, instead of ice nucleation onsets, full INP spectra or ice-nucleation active site densities should be reported. As explained above, this is not achievable with emulsions, which always lead to polydisperse droplet distributions. The method of choice to obtain monodisperse droplet populations of small enough size to avoid ice nucleation on impurities is in our opinion microfluidics.

*On a more specific aspect of the paper, the arguments in Section 4.5 should perhaps consider more than size as influencing nucleating ability of a site. Ice nucleation literature abounds in discussions about the roles of unique sites (crystal steps, specific molecular arrangements, etc.) versus the general properties of a substrate surface. This paper considers the size necessary for embryo formation on the basis of CNT, thus implicitly assuming that average surface properties form the criteria for nucleation. It would increase the value of the paper if the authors included some discussion about the possible role of unique sites.*

Our discussion of the IN activity of smectites concentrates on size because smectite platelets are so thin that size matters for nucleation sites. We agree that size is not the only criterion for a nucleation site. Section 4.5, titled "Connection between particle thickness and IN activity" concentrates specifically on size. Therefore, in this section, we neglect the other relevant prerequisites for ice nucleation. We implicitly acknowledge the relevance of other criteria when we write in the very first line of Section 4.5 that the platelet thickness is "a limiting factor" and not "the limiting factor".

In Section 5, we mention the relevance of surface functionalization as another relevant criterion, as we write in the second paragraph: "Furthermore, as the edges are densely hydroxylated, they fulfill a prerequisite for ice nucleation. Hydroxylated aluminosilicate surfaces are a feature that clay minerals have in common with feldspars, whose IN activity also increases in the presence of ammonia and ammonium solutions. Thus, a sufficiently large area equipped with hydroxy groups that are available for hydrogen bonding to water molecules seems key for ice nucleation by aluminosilicates in general."

Following the suggestion of Gabor Vali, we add to this: "Besides the precise location of hydroxyl functionalities within the IN active area, additional topographical properties of the edge surface might also influence the quality of the nucleation sites."

**References**

Kaufmann, L., Marcolli, C., Hofer, J., Pinti, V., Hoyle, C. R., and Peter, T.: Ice nucleation efficiency of natural dust samples in the immersion mode, Atmos. Chem. Phys., 16, 11177–11206, doi:10.5194/acp-16-11177-2016, 2016.

Marcolli, C., Gedamke, S., Peter, T., and Zobrist, B.: Efficiency of immersion mode ice nucleation on surrogates of mineral dust, Atmos. Chem. Phys., 7, 5081–5091, doi:10.5194/acp-7-5081-2007, 2007.

.

We thank the reviewer for the constructive comments. We reproduce his/her comments in *blue* and our responses in black.

*I feel the authors have adequately addressed my concerns. However, I would still argue that the way the authors treat the DSC curve analysis is rather rough and perhaps more quantitative data could emerge if better curve fitting algorithms were employed. Although we have not made the same suggestions I see from the Vali response that he had similar ideas. Moreover, it seems in the response to his point that a peak in the differential spectrum of the DSC curve would correspond precisely with the location of the steepest slope and that might be another useful marker seemed to be a bit misinterpreted. This is not peak temperature, as the response seems to narrow in on. Finally because the shapes of the curves (i.e., the tails) will not always be the same for the het and hom cases I do not entirely buy the argument that the chosen method of integration would not be significantly outperformed by a doublet deconvolution, and thus perhaps yield more insight. All of the singly peaked DSC curves are quite symmetrical, why would we not expect the same of others? Thus knowing one tail should allow robust fitting of the other (the part convoluted with the other freezing behavior).*

As we have explained in our responses to Gabor Vali, a curve fitting and deconvolution would not inform us about the nucleation events. Repeating here a part of the response we have given to Gabor Vali's comments, the heat signal just informs us about the heat release during freezing, which is delayed with respect to ice nucleation and droplet freezing as discussed in Marcolli et al. (2007). Even if nucleation resulted in complete, instantaneous freezing, we would not be able to derive nucleation events from the DSC signal, because the delay in heat transfer is substantial. The analysis in Marcolli et al. (2007) indicates that in our DSC instrument, the heat release of each droplet follows an exponential decay law with a time constant of 11 s. Most importantly, the heat retention transiently increases the DSC sample temperature, resulting in a temporary deviation from the nominal cooling rate. Depending on the intensity of the heat release, nucleation might be almost complete when the heat signal peaks. Therefore, the onset is the best-defined temperature to reference a heat signal.

Moreover, in emulsion freezing experiments, the INP surface or mass per droplet is subject to large uncertainties because of the polydispersity of the emulsion droplets as discussed in the appendix of Kaufmann et al. (2016). Because of the uncertainties in nucleation temperatures and number of INPs per droplet, emulsion freezing experiments performed with DSC are not suited to derive INP spectra or ice-nucleation active site densities. The method of choice to obtain monodisperse droplet populations of small enough size to avoid ice nucleation on impurities is in our opinion microfluidics.

*This said, the author's larger point is that this information leads only to relative weighting and this is only one of a few barriers in the technique that prevent quantified nucleation information from emerging. Rather the point is more to give insight into what promotes freezing earlier versus later.*

*Itemized Scientific and Editorial Comments from the resubmitted tracked changes manuscript version:*

*Specific Suggestions by Page and Line Number (page, line):*

*• (11,293) should read....and are also shown in....*

Corrected.

*• (11,294) Rigorously ~ means similar to, as in the same order of magnitude, whereas ≈ means approximately. Here the later should be used given the range is less than 1 order of magnitude.*

Corrected.

*• (11,300) should read....In the case of emulsions formed with pure water....*

Corrected.

*• (Figure 1, caption) should read.....For references purposes a DSC....for the samples.*

Corrected.

*• (22,487) The 2 sentences beginning, "Though (001)," and ending..."(Ren et al., 2020)" are difficult to read and include dangling clauses etc. I would suggest rephrasing to improve clarity.*

The sentence has been rephrased for clarity.

*• (30,673) should read....Only a few....*

Corrected.

*• (30,677) should read...started with dry particles*

Corrected.

*• (30,689) should read......atomic configurations for ....*

Corrected.

*• (30,690) should read...in simulations to obtain....*

Corrected.

*A general comment: at times the usage of "in immersion freezing mode" seems like it would better written ....in the immersion freezing mode.*

Corrected at several places in the revised manuscript.

*Although willing to review another edition of this manuscript, I do not see any acute need. I can recommend the manuscript for publication when these technical corrections are made and the editor is happy.*

**References**

Kaufmann, L., Marcolli, C., Hofer, J., Pinti, V., Hoyle, C. R., and Peter, T.: Ice nucleation efficiency of natural dust samples in the immersion mode, Atmos. Chem. Phys., 16, 11177–11206, doi:10.5194/acp-16-11177-2016, 2016.

Marcolli, C., Gedamke, S., Peter, T., and Zobrist, B.: Efficiency of immersion mode ice nucleation on surrogates of mineral dust, Atmos. Chem. Phys., 7, 5081–5091, doi:10.5194/acp-7-5081-2007, 2007.